# An Evaluation of the EnKF vs. EnOI and the Assimilation of SMAP, SMOS and ESA CCI Soil Moisture Data over the Contiguous US

**Jostein Blyverket [1,2,\*]** , **Paul D. Hamer [1]** , **Laurent Bertino [3]** , **Clément Albergel [4]** , **David Fairbairn [5]** and **William A. Lahoz [1]**

[1] NILU—Norwegian Institute for Air Research, INBY, Instituttveien 18, 2007 Kjeller, Norway; pdh@nilu.no (P.D.H.); wal@nilu.no (W.A.L.)

[2] Geophysical Institute, University of Bergen UiB, Allegaten 70, 5020 Bergen, Norway

[3] Nansen Environmental and Remote Sensing Center, 5020 Bergen, Norway; laurent.bertino@nersc.no

[4] CNRM—Université de Toulouse, Météo-France, CNRS, 31057 Toulouse, France; clement.albergel@meteo.fr

[5] European Centre for Medium-range Weather Forecasts (ECMWF), Reading RG2 9AX, UK; David.Fairbairn@ecmwf.int

\* Correspondence: jb@nilu.no

**Abstract:** A number of studies have shown that assimilation of satellite derived soil moisture using the ensemble Kalman Filter (EnKF) can improve soil moisture estimates, particularly for the surface zone. However, the EnKF is computationally expensive since an ensemble of model integrations have to be propagated forward in time. Here, assimilating satellite soil moisture data from the Soil Moisture Active Passive (SMAP) mission, we compare the EnKF with the computationally cheaper ensemble Optimal Interpolation (EnOI) method over the contiguous United States (CONUS). The background error–covariance in the EnOI is sampled in two ways: (i) by using the stochastic spread from an ensemble open-loop run, and (ii) sampling from the model spinup climatology. Our results indicate that the EnKF is only marginally superior to one version of the EnOI. Furthermore, the assimilation of SMAP data using the EnKF and EnOI is found to improve the surface zone correlation with in situ observations at a 95% significance level. The EnKF assimilation of SMAP data is also found to improve root-zone correlation with independent in situ data at the same significance level; however this improvement is dependent on which in situ network we are validating against. We evaluate how the quality of the atmospheric forcing affects the analysis results by prescribing the land surface data assimilation system with either observation corrected or model derived precipitation. Surface zone correlation skill increases for the analysis using both the corrected and model derived precipitation, but only the latter shows an improvement at the 95% significance level. The study also suggests that assimilation of satellite derived surface soil moisture using the EnOI can correct random errors in the atmospheric forcing and give an analysed surface soil moisture close to that of an open-loop run using observation derived precipitation. Importantly, this shows that estimates of soil moisture could be improved using a combination of assimilating SMAP using the computationally cheap EnOI while using model derived precipitation as forcing. Finally, we assimilate three different Level-2 satellite derived soil moisture products from the European Space Agency Climate Change Initiative (ESA CCI), SMAP and SMOS (Soil Moisture and Ocean Salinity) using the EnOI, and then compare the relative performance of the three resulting analyses against in situ soil moisture observations. In this comparison, we find that all three analyses offer improvements over an open-loop run when comparing to in situ observations. The assimilation of SMAP data is found to perform marginally better than the assimilation of SMOS data, while assimilation of the ESA CCI data shows the smallest improvement of the three analysis products.

**Keywords:** land data assimilation; EnKF; EnOI; SMAP; SMOS; ESA CCI for soil moisture

---

## 1. Introduction

Accurate knowledge of surface and root-zone soil moisture is important since it constrains the energy and water exchanges across the land–atmosphere interface. Soil moisture (SM) controls the surface infiltration vs. runoff process, water available for the biosphere through plant transpiration, and partitioning of incoming energy into sensible and latent heat fluxes [1,2]. Several studies have investigated if initialization of SM could help the skill of seasonal hydrological predictions [3–5], and also in which parts of the world such seasonal predictions are sensitive to the initial hydrological conditions [6]. In such regions, knowledge about the initial conditions of SM can provide a skilful prediction of subsequent SM conditions using historical atmospheric forcing or seasonal climate forecasts. One way of improving the SM state is through data assimilation (DA); this approach combines information from satellite observations with model data, using a mathematical framework, see, for example, [7–14].

Microwave satellite instruments allow us to monitor and map SM at the global scale, and provide a wealth of information that can be used to inform and improve Earth surface modelling [15]. However, satellite remote sensing of SM has limitations because of spatial and temporal gaps in the resulting data product. For example, current satellite instruments used to derive SM are operating in the X, C and L-band, which are mostly sensitive to SM in the upper soil layer (0–5 cm), and not the root-zone layer [16]. Spatial gaps are also found in the horizontal direction because not all parts of the globe will be covered every day, while temporal gaps are related to the revisit time of the satellites.

By merging land surface modelling and satellite observations using DA, we can create a product which combines the spatial and temporal coverage of a land surface model (LSM) with satellite derived surface SM (SSM). The LSM controls the partitioning of rainfall into runoff, evapotranspiration, drainage and soil moisture. Satellite observations can compensate for model parameterizations and precipitation events not described in the atmospheric forcing. For example, in Koster et al. [17], the authors found that, by assimilating and utilizing SM derived from the Soil Moisture Active Passive (SMAP) satellite, they were able to improve the model representation of dry-down events. By knowing the errors of the model and the observations, DA has the potential to create a superior product over and above either model or satellite SM alone. For instance, this approach has been taken in the development of the SMAP Level-4 product, where modelled SM is converted to brightness temperature and optimized towards the observed brightness temperature measured by SMAP [18].

One of the most applied and successful methods in land DA is the ensemble Kalman Filter (EnKF) [19]. This method solves the Kalman filter equations, while sampling the background error-covariance from an ensemble of model trajectories. The EnKF allows for flow dependent background errors and also gives the uncertainty of the analysis in the form of the ensemble spread. The input errors are set by perturbing the atmospheric forcing and/or the state variables and/or model parameters. How to correctly specify these errors on a larger domain is not trivial, and it is expected that an improved spatial description of model and observation errors will benefit land surface DA [12]. One of the major caveats with the EnKF is the high computational cost of the forward integration of an ensemble of model states [20] and references therein. This often make the EnKF unattractive for operational applications over larger domains.

To keep the computational cost at a minimum while still retaining some of the benefits from the EnKF, we suggest to use the ensemble Optimal Interpolation (EnOI) method [21,22]. The EnOI solves the same Kalman Filter equations as the EnKF; however, it relies on a prescribed background error-covariance. Land DA systems using the EnKF often do the ensemble forward integration twice, once for normalizing the satellite SM data to the model and the second integration for doing the analysis. Here, we suggest two methods for describing the background error-covariance in an EnOI

system: (i) using the ensemble open-loop forecast, which can be obtained from the model ensemble used to normalize the satellite SM data, and (ii) using a climatological background error-covariance from the model spinup. The major benefit with method (i) is that we only need to do the ensemble forward integration once, while, for method (ii), no ensemble integration is needed at all. More details about these methods are presented in Section 3.4.

The relative improvements of SM estimates between the analysis in a land DA system and the open-loop (i.e., no DA) run are affected by the quality of the atmospheric forcing used to drive the land surface model [23,24], whereby the usage of observation corrected or model derived precipitation tends to result in lower or higher impact of the land DA, respectively. A model derived precipitation dataset is expected to e.g., have more missed precipitation events, or represent precipitation events that never occurred, and these random errors in the precipitation field can be compensated for by land DA of satellite derived surface SM. In our study, we utilize two different atmospheric forcing datasets, (i) the Modern Era Retrospective-analysis for Research and Applications version 2 data (MERRA-2) from the Global Modeling and Assimilation Office (GMAO) [25] and (ii) data from the North American Land Data Assimilation System version 2 (NLDAS-2) [26]. By applying the observation corrected precipitation product from the NLDAS-2 dataset, and the model derived precipitation from the MERRA-2 dataset, we are able to evaluate how these atmospheric forcing datasets affect our land DA analysis results.

The main goal of this paper is to assess the EnOI versus the EnKF for soil moisture DA. The skill of the two filters is evaluated using DA diagnostics and in situ SM measurements. The DA diagnostics checks that the assumptions underlying the Kalman Filter equations are met, e.g., the whiteness of the observation-minus-forecast (O-F) residuals, and that the expected model and observation errors are close to the actual errors [27]. We do the model experiments over the contiguous United States (CONUS) because this region incorporates a wide range of biomes with different climatic conditions, while at the same time it has numerous in situ SM stations to use for validation. In situ observations act as an independent reference for testing the skill of the analysis vs the open-loop. Furthermore, the comparison to in situ SM observations is utilized to: (i) evaluate the impact of the quality of the atmospheric forcing on the land DA results, (ii) assess whether a land DA analysis driven by model derived precipitation can have comparable skill to an open-loop run using observation corrected precipitation, and (iii) assess the skill of the land DA analysis using three different SM satellite products from the European Space Agency Climate Change Initiative (ESA CCI), Soil Moisture and Ocean Salinity (SMOS) and SMAP, respectively. Section 2 presents the different satellite SM products, in situ observations and model auxiliary data. In Section 3, we describe the bias correction, modelling system and data assimilation algorithms. Section 4 describes the experiments and validation metrics. Section 5 presents the results.

## 2. Data

### 2.1. Satellite Derived SSM

#### 2.1.1. ESA CCI ACTIVE and PASSIVE SSM Product

The ESA CCI for soil moisture project has been ongoing since 2011; the goal of the project is to create a global, consistent SSM product using active and passive satellite sensors [28–31]. As of 18 January 2019, the dataset spans from 1978 until 30 June 2018. The ESA CCI SSM product is divided into three different datasets. First, the ACTIVE product, which consists of daily SSM values from active sensors, using backscatter as a measure of SSM. Second, the PASSIVE product, which uses passively observed brightness temperature as a measure of SSM. Finally, the COMBINED product, fusing the ACTIVE and PASSIVE product into one, for improved spatial and temporal coverage.

The ACTIVE SSM retrieval product is based on the change detection method developed at the Vienna University of Technology (TU Wien) [32–34]. The change detection method is derived for C-band scatterometers. Surface soil moisture is retrieved directly using scatterometer measurements

without the need for an iterative adjustment process. It is assumed that the relationship between the SSM and the backscatter coefficient $\sigma^0$ is linear. For the time-period 31 March 2015–31 December 2016 used in our study, the ACTIVE SSM observations stem from the METOP-A and B satellites (ASCAT-A and ASCAT-B).

The PASSIVE SSM dataset is derived using the land surface brightness temperature (Tb) in the microwave range, which is related to the soil dielectric constant and hence the SSM content. The conversion from observed Tb to SSM values is handled by the Land Parameter Retrieval Model (LPRM) [35]. The LPRM is a radiative transfer model that simultaneously retrieves vegetation density, SSM and surface temperature.

Several studies have assimilated the different ESA CCI products, e.g., [13,36]. Draper and Reichle [37] found that the largest improvements in SSM were seen when assimilating both the active and passive data, compared to assimilating active/passive data only. For this reason, we simultaneously assimilate the separate ACTIVE and PASSIVE products, which are given in the range of the change detection algorithm and the land parameter retrieval model, respectively. The different overpass times of the satellites in the ACTIVE and PASSIVE products allow for assimilation of both products, without any further pre-processing. In this work, we use the ESA CCI SSM v04.2, which became public in January 2018. The ESA CCI ACTIVE and PASSIVE products are delivered on a regular longitude/latitude grid. We remap the ESA CCI observations to the model grid (with a resolution of 25 km) using a nearest neighbour approach. This ensures that the observed SSM values are not smoothed values from an interpolation scheme. Further pre-processing of the satellite data is explained in Section 3.1. More information about the ESA CCI SSM product, including quality assessment and range of uses is available in Refs. [29,30].

### 2.1.2. SMAP Level-2 SSM

The Soil Moisture Active Passive (SMAP) satellite measures passive microwave emissions from the Earth's surface [38]. It was launched in 2015 by NASA, carrying a passive instrument (L-band radiometer) and an active instrument (L-band radar). The active instrument failed shortly after launch, hence we only use data from the radiometer. The satellite derived SSM is an average over the whole footprint area, which is around 40 km for L-band radiometers. The SMAP Level-2 (L2) SSM product is extracted from the National Snow and Ice Data Center; we use the v5 R16010 product covering the period 31 March 2015–31 December 2016. We only use SMAP-L2 data where the SSM uncertainty is less than $0.1 \, \mathrm{m^3 m^{-3}}$, the SSM is in a realistic range ($0.0–0.6 \, \mathrm{m^3 m^{-3}}$), and where the LSM observation layer (layer-2) has a temperature above 2 °C [12]. The SMAP-L2 swath data are gridded to the LSM grid using a nearest neighbour approach.

### 2.1.3. SMOS Level-2 SSM

The Soil Moisture and Ocean Salinity (SMOS) satellite, was launched in 2010 by ESA, and similarly to SMAP, SMOS uses a radiometer instrument operating in the L-band [39,40]. In contrast to the SMAP satellite, which has a constant observation incidence angle, SMOS measures brightness temperature for a range of observation incidence angles. This angular information can be used to separate soil and vegetation signals over land [41]. The SMOS-L2 SSM data are extracted from the SMUDP2 v650 reprocessed product obtained from the ESA SMOS dissemination service. To match the experiment period, we use SMOS-L2 data spanning from 31 March 2015–31 December 2016, which is the overlapping time-period between the SMAP-L2, SMOS-L2 and the ESA CCI v4.2 products. The quality controls applied before bias correction consist of (i) removing observations outside a realistic SM range ($0.0–0.6 \, \mathrm{m^3 m^{-3}}$), (ii) choosing SM values with uncertainties less than $0.1 \, \mathrm{m^3 m^{-3}}$, (iii) choosing SM values for instances where the radio frequency interference (RFI) probability is less than 0.3, and (iv) choosing SM values where the retrieval flag is not raised for snow and ice. Finally, the observations are discarded if the soil temperature in the LSM is below 2 °C. The SMOS-L2 swath values are gridded to the model grid using the same approach as for the SMAP-L2 SSM product.

## 2.2. In Situ Data

The high number of in situ stations over the United States allows us to compare the land DA results to independent reference data. This allows for a more robust comparison and validation of the analysis results. We use data from the International Soil Moisture Network (ISMN) [42,43], and, more specifically, data from the Soil Climate Analysis Network (SCAN) [44] and the U.S. Climate Reference Network (USCRN) [45,46]. A promising way to validate the land surface data assimilation system is by using cosmic-ray neutron measurements from the COsmic-ray Soil Moisture Observing System (COSMOS) [47,48]. This network is not applied in our study because we wanted to facilitate the comparison of average skill metrics across the SCAN and USCRN networks as done in other land DA studies. There are several limitations when validating satellite and model SM data using sparse in situ networks. These limitations include: (i) spatial mismatch (representativeness error) between what the station (i.e., a point) measures and what the model and satellite measure (i.e., domain averaged SM), and (ii) in situ station measurements have instrumental errors. The in situ data are pre-processed following Reichle et al. [49], this means that: (i) we only use data flagged as good in the ISMN data product, (ii) all values are discarded if the measured soil temperature at that level is below $2\,°C$, and (iii) we only use SM values in the range between 0 and $\theta_{saturation}$ for that station. The $\theta_{saturation}$ information for a given station is obtained from the ISMN auxiliary data product. After this quality control of the data, we create a temporal average using the preceeding and succeeding hourly values of a 3 h window. Surface zone soil moisture (sfzsm) and root-zone soil moisture (rzsm) are calculated for the individual in situ stations following the approach in Ref. [12], and excluding the in situ SM sensor at 0.5 m depth following [50].

## 2.3. Atmospheric Forcing

We use atmospheric forcing from NLDAS-2 [26]; this dataset provides hourly input of precipitation (PRECIP), air temperature (Tair), specific humidity (SH), longwave downward (LW) and shortwave downward radiation (SW). The NLDAS-2 dataset has observation corrected precipitation and also bias corrections for incoming SW radiation; this means that the forcing dataset error will be small compared to the errors associated with, e.g., a forecast product. The NLDAS-2 forcing is upscaled from the native 12.5 km grid to the model 25 km grid using bilinear interpolation. The NLDAS-2 dataset is a high-resolution dataset focused on North America, which allows it to ingest high-quality precipitation observations at a fine scale.

The second atmospheric forcing dataset is from the Modern Era Retrospective-analysis for Research and Applications version 2 (MERRA-2) [25]. The MERRA-2 product is a state-of-the-art global atmospheric reanalysis with a resolution of $0.5° \times 0.625°$. From MERRA-2, we utilize hourly input of PRECIP, Tair, SH, LW and SW. To evaluate the impact of the atmospheric forcing on the land DA analysis, we apply the MERRA-2 PRECIP without any observation corrections. The MERRA-2 product was downloaded from the Goddard Earth Sciences Data Information Services Center (GES DISC). We use bilinear interpolation to remap the MERRA-2 product from the native grid to the LSM grid.

## 3. Methods

### 3.1. Bias Correction of the ESA CCI ACTIVE and PASSIVE, SMAP-L2 and SMOS-L2 SSM Products

In sequential DA algorithms, it is assumed that there is no bias between the model and the observations. Sequential DA methods only correct random errors, hence systematic errors in the land DA system need to be removed. In land DA, this is often done by matching the statistical moments of the observations to that of the model, e.g., [51,52]. The ACTIVE and PASSIVE products are delivered as daily products, and they are a composite of different satellite overpasses at a given date. To avoid additional bias between the model and observations, the timing of the forecast (model) and observation (satellite retrieval) need to coincide as much as possible. Following De Lannoy and Reichle [12], we

use a temporal gap of $\pm 1.5$ h centred at 00, 03..., and 21 UTC; satellite overpasses within this time window are mapped to the centre of the time window.

The range of the ACTIVE product is given in percent, 0–100%. The ACTIVE data are converted to volumetric SM using the model minimum and maximum SM for a given grid-cell. For the conversion to volumetric SM, we only use model timestamps where we also have observational data. The PASSIVE product is given in volumetric SM and does not need any additional unit conversion.

To handle the bias between the model and observations, we rescale the observations using cumulative distribution function matching (CDF-matching) as in [52,53]. We note that different bias correction methods could have an impact on the DA results as seen in, e.g., [54,55]. In this study, we assume that the biases between the satellite derived SSM and modelled SSM are close to being stationary [12], which means that we can use a lumped CDF-approach. If the biases are seasonally varying, this assumption could lead to remaining biases and dampening of short-term variability in the satellite derived SSM, as pointed out by, e.g., [9].

On the technical side, the CDF-matching is performed for each individual grid-cell independently, and requires that a given grid-cell has more than 200 observations over the whole time-period. Otherwise, all the values at that particular grid-cell are discarded. The CDF-matching works by ranking the observations and model values for an individual grid-cell at a specific time e.g., 0900 UTC. By taking the difference between the modelled and observed ranked datasets and then fitting a 5th order polynomial to these points, we can find the new observed value by adding this difference to the old observed value. The CDF-matched observations will then have a CDF which is matched with the CDF of the model for that grid-cell and time of the day.

*3.2. Modelling System*

In our study, we use the SURFEX v.8.0 (SURFace EXternalisée [56]) modelling framework. Within the SURFEX land surface modelling platform, we use the Interaction between Soil Biosphere and Atmosphere diffusion scheme (ISBA-DF) [57–59]. Vertical transport of water is solved using the mixed form of Richard's equations while the soil temperature is solved using the one-dimensional Fourier law. The ISBA model also includes soil freezing [60] and an explicit snow scheme [61]. Fourteen vertical layers are used over a depth of 12 m depending on grid-cell characteristics. The depth of the different layers are the same as in [13]. Layer one is a skin layer, we use layer two (between 0.01 and 0.04 m) as the model equivalent of the observation layer. The time-step for the LSM is set to 30 min, output is saved at 3 h intervals. In this work, we only use one sub-grid patch where the parameters for the mass and energy balance are aggregated from the different land covers within the grid-cell. More information about the model can be found in Decharme et al. [59].

Land cover is extracted from ECOCLIMAP, a global database at 1 km resolution for land surface parameters [62]. For nature grid-cells in the model, surface parameters are computed using the fraction of 12 vegetation types from the 1 km resolution land cover map. The leaf area index (LAI) is derived from the ECOCLIMAP database and given as a climatology for each calendar month. The clay and sand fractions are extracted from the Harmonized World Soil Database (HWSD) [63]. In SURFEX, several soil parameters are derived from the clay and sand fraction using formulas from Noilhan and Lacarrere [64]. The hydraulic conductivity and soil water potential are related to the liquid soil water content through the Clapp and Hornberger relations. Grid-cell orography is computed in SURFEX by aggregating the GTOPO 1 km resolution elevation dataset to the 25 km model grid.

*3.3. Data Assimilation Using the Ensemble Kalman Filter*

There are several different versions of the EnKF; in this work, we use the Ensemble Square Root Filter (ESRF) from Sakov and Oke [65]. This algorithm is applied both in the EnKF and the EnOI methods, for simplicity we will refer to the ESRF as the EnKF in the rest of this paper. The EnKF interface with SURFEX in this work has been developed at the Norwegian Institute for Air Research (NILU) and it is different from the EnKF developed at the National Centre for Meteorological

Research (CRNM) [20,66]. The state vector **x** consists of the top four model layers, i.e., $\theta_1$ (0.001–0.01 m), $\theta_2$ (0.01–0.04 m), $\theta_3$ (0.04–0.10 m) and $\theta_4$ (0.10–0.20 m) for all of the $N$ ensemble members. Owing to spurious correlations found between the model equivalent observation layer (layer-2) and layers 5 to 14, we choose to only update the top four model layers. Hence, the root-zone is not explicitly updated in the DA analysis. An observation vector **y** holds the satellite observation ($\theta_{\mathrm{obs}}$); the EnKF combines an ensemble of model states ($\mathbf{x}_N$) with the observations **y**. For both the EnKF and EnOI, we apply a three-hourly assimilation window, and this is to ensure that the model forecast SSM is as close as possible to the satellite derived SSM. The EnKF analysis equation is written as:

$$\mathbf{x}_i^a = \mathbf{x}_i^f + \mathbf{K}(\mathbf{y} - \mathbf{H}\mathbf{x}_i^f),$$

where $\mathbf{x}_i^a$ is the analysis for ensemble member $i$; $\mathbf{x}_i^f$ is the forecast state; **y** is the vector of observations and **H** linearly maps the state vector to observation space. The Kalman gain **K** is given as:

$$\mathbf{K} = \mathbf{B}^f\mathbf{H}^T(\mathbf{H}\mathbf{B}^f\mathbf{H}^T + \mathbf{R})^{-1}$$

in which **R** is the observation error-covariance matrix and $\mathbf{B}^f$ is the forecast error-covariance matrix. The analysis error-covariance is updated according to:

$$\mathbf{B}^a = (\mathbf{I} - \mathbf{K}\mathbf{H})\mathbf{B}^f,$$

for the EnKF, where **I** is the identity matrix. In the EnKF, the model error-covariance matrix $\mathbf{B}^f$ is stored and manipulated implicitly via an ensemble $\mathbf{x}_N$ of $N$ model states, and is given by:

$$\mathbf{B}^f = \frac{1}{N-1}\sum_{i=1}^{N}(\mathbf{x}_i - \mathbf{x})(\mathbf{x}_i - \mathbf{x})^T = \frac{1}{N-1}\mathbf{A}\mathbf{A}^T,$$

where **x** is the ensemble mean defined as

$$\mathbf{x} = \frac{1}{N}\sum_i^N \mathbf{x}_i.$$

In the above equations, **A** represents the deviations from the ensemble mean. The EnKF used here is currently a 1-D scheme with independent soil columns in the $x$- and $y$-directions. The observation operator is set to: $\mathbf{H} = (0, 1, 0, 0)$, so that it only selects the cross-correlations between the observation layer and layers 1,3 and 4. We apply 12 ensemble members for the EnKF and the EnOI, this choice is based on: (i) the state vector is relatively small (four variables), (ii) other studies have found that this number is sufficient and there is little to gain when increasing it further [37,67,68], and (iii) as an effort to minimize the computational cost.

### 3.4. Ensemble Optimal Interpolation (EnOI)

In the EnKF equations above, the background error-covariance is modified in the analysis step, and this allows for a flow dependent background error. However, this comes with an added computational cost as we need $N$ model integrations ($N$ being the number of ensembles). Here we argue that an ensemble Optimal Interpolation (or statistical interpolation) method could perform well in a land surface DA system [21,22]. The main advantage of the EnOI is the low computational cost since we only need one ensemble member. This is especially useful for operational systems and for reanalyses where we run an ensemble open-loop for the period of interest to normalize the satellite data (as done here). Then, we can use the background error-covariance from the ensemble open-loop for "free" in the EnOI. It is also worth noting that, when moving towards coupled land–atmosphere DA systems, the EnOI could be even more beneficial since we only need one model integration of the coupled land–atmosphere system. The caveat of the EnOI is how to determine a realistic background error;

in this study, we use the forecast spread from the ensemble open-loop run, and a climatologically sampled **A** of the background error-covariance. The EnOI equations are given as:

$$\mathbf{x}^a = \mathbf{x}^f + \mathbf{K}'(\mathbf{y} - \mathbf{H}\mathbf{x}^f),$$

where

$$\mathbf{K}' = \mathbf{B}'^f \mathbf{H}^T (\mathbf{H}\mathbf{B}'^f \mathbf{H}^T + \mathbf{R})^{-1}$$

and the background error-covariance is:

$$\mathbf{B}'^f = \frac{1}{N-1} \sum_{i=1}^{N} (\mathbf{x}'_i - \mathbf{x}')(\mathbf{x}'_i - \mathbf{x}')^T = \frac{1}{N-1} \mathbf{A}'\mathbf{A}'^T,$$

where $\mathbf{A}'$ is the deviation from the ensemble mean of the prescribed model spread. The number of ensembles $N$ used to calculate the anomalies is 12 for the EnOI. In the EnOI, we only update a single model run instead of updating the whole ensemble as in the EnKF. This means that there is no ensemble spread after the analysis and hence no information about the uncertainty.

The EnOI with a climatological background error (EnOI-Clim) solves the same equations as outlined above. The differences between the EnOI and EnOI-Clim arise from the way the ensemble anomalies are computed. We calculate the EnOI-Clim anomalies by sampling from the model spinup (single model run). First, we create a climatology for each of the assimilation windows, and then $N$ values from that time of the day are subtracted from the climatology, and hence creating the ensemble anomalies. Following Counillon and Bertino [69], we introduce a scaling factor $\alpha(\in (0,1])$, which in this work is set to $\alpha = 0.3$ to dampen the ensemble variance. For the EnOI-Clim, we choose $N = 122$ (days) to represent the number of ensembles. Our ensemble was created using March, April and May data from the years 2012, 2013 and 2014; we choose to use a stationary sampled climatology in the representation of the ensemble anomalies. Later work will address whether a seasonally varying climatology would have a positive impact on the EnOI-Clim analysis results. The downside with the EnOI-Clim method is that it will not represent the errors of the day; however, it is expected that, over time, useful information will be transferred from the satellite observation and into the model via this analysis.

### 3.5. Model and Observation Errors

Models are simplified representations of the real world and have different sources of errors. For a land surface model, some of these errors are: (i) error of representativeness, (ii) error in model parameters/parameterization of physical processes, and (iii) external errors such as atmospheric forcing, land cover classification and sand and clay fraction. These different errors can in theory be represented in the DA system. Here we represent errors in the forcing (by perturbing the different forcing fields) and model parameter errors (by perturbing the model parameters, which partially encompass the sub-grid scale representativeness errors).

The perturbations done to the atmospheric forcing follow what is done in [23,49,67,70]. Time-series of cross-correlated forcing fields are generated using an autoregressive lag-1 model (AR(1)). These perturbations in the atmospheric forcing allow for an ensemble of model runs, where the spread represents the model uncertainty. Precipitation and shortwave radiation have a lower bound of zero, the perturbations of these variables are therefore multiplicative. Perturbations in longwave radiation are additive. To avoid bias in the forcing files, the perturbations are constrained to zero for the additive variables and to one for the multiplicative variables. The pseudorandom field $q_t$ is given by:

$$q_t = \alpha q_{t-1} + \beta w_t,$$

where $\alpha = \frac{\Delta t}{\tau}$ and $\beta = \sqrt{(1 - \alpha^2)}$ where $\tau = 24\,\text{h}$ and $\Delta t = 1\,\text{h}$. The $w_t$ is a normally distributed random field. To ensure physical consistency in the perturbation parameters (e.g., increase in longwave radiation gives decrease in shortwave radiation), we impose cross-correlations on the pseudorandom fields using the correlations given in Table 1. The ensemble is kept unbiased centred at the original variable, which implies that the original forcing variable is an ensemble member and at the same time is the ensemble mean. Note that, since the land surface model is nonlinear, this needs not to be true for the prognostic model variables [71].

Errors in the model parameters are based on perturbation of the clay and sand fractions. In SURFEX, the clay and sand fraction acts as proxy variables for the saturated, wilting point and field capacity volumetric water content. Thus, these changes in the sand and clay fractions will ultimately lead to different model physics for the different ensemble members. Since these fractions are bounded by 0 and 1, we use a Logit-normal transformation following the approach in [72]. The parameters are first logit transformed; then, we add $N$ ensemble members of Gaussian white noise to the logit transformed variable before we do the inverse transformation. This is done for each individual model grid-cell using a uniform spatial random variable and a standard deviation of $\pm 10\%$.

**Table 1.** Perturbation parameters for the atmospheric variables precipitation (PRECIP), shortwave (SW) and longwave (LW) downward radiation and the model parameters (clay and sand fractions). The perturbations are multiplicative (M), additive (A) and Logit-normal. The cross-correlation between the variables are given in the Cross-corr columns.

| Variable | Type | Std Dev | Cross-Corr with Perturbations | | |
| --- | --- | --- | --- | --- | --- |
| | | | PRECIP | SW | LW |
| PRECIP | M | 0.5 | 1 | −0.8 | 0.5 |
| SW | M | 0.3 | −0.8 | 1 | −0.5 |
| LW | A | $30\,\text{W/m}^2$ | 0.5 | −0.5 | 1 |
| clay | Logit normal | 0.1 | - | - | - |
| sand | Logit normal | 0.1 | - | - | - |

The setting of observation errors is based on the assumption of a linear relationship between the dynamic range of SM values and the errors [66,73]. This relationship is given as $(\theta_{\text{fc}} - \theta_{\text{wilt}})$, where $\theta_{\text{fc}}$ is the field capacity and $\theta_{\text{wilt}}$ is the wilting point, they both depend on the soil texture and vegetation type for each grid-cell. This relationship is scaled with a dimensionless scaling coefficient $\lambda^{\text{o}}$, resulting in an observation error given by: $\lambda^{\text{o}}(\theta_{\text{fc}} - \theta_{\text{wilt}})\,\text{m}^3\text{m}^{-3}$. Setting $\lambda^{\text{o}} = 0.3$, results in a spatial average observation error of $0.026\,\text{m}^3\text{m}^{-3}$, which is slightly larger than the spatial average reported by De Lannoy and Reichle [12] in their study over CONUS. We assume a diagonal **R** matrix with no cross-correlated observation errors. In this work, we choose to use the same observation error for the three different satellite products. This is because the current study is on of the first that compares DA analyses of the three Level-2 soil moisture products. Hence, we would like to compare them on equal ground. Further tuning of the observation error would have to be addressed in future work.

## 4. Experiments and Skill Metrics

The land surface DA experiments are designed as follows. Perturbations are applied to the model parameters (clay and sand fractions), in order to represent model parameter errors. The atmospheric forcing is perturbed according to the process defined in Section 3. This creates an ensemble of model configurations, and the ensembles are propagated 3 h forward in time by the ISBA LSM. Given the model background error from the ensemble and the observational error, the EnKF weights each individual ensemble and updates the state vector. The updated state vector is the initial condition for the next model integration. Given correct specification of background and observation errors the updated state vector should not be worse and should in general be superior to the model or satellite

alone. Information from the observation layer (layer two in the state vector) is propagated downward in the analysis update by **K** and subsequently by the model integration. In the EnOI experiments, we use either the ensemble open-loop model integration or the climatological sampled background covariance matrix in the analysis step—see Figure 1. The full list of model experiments is found in Table 2.

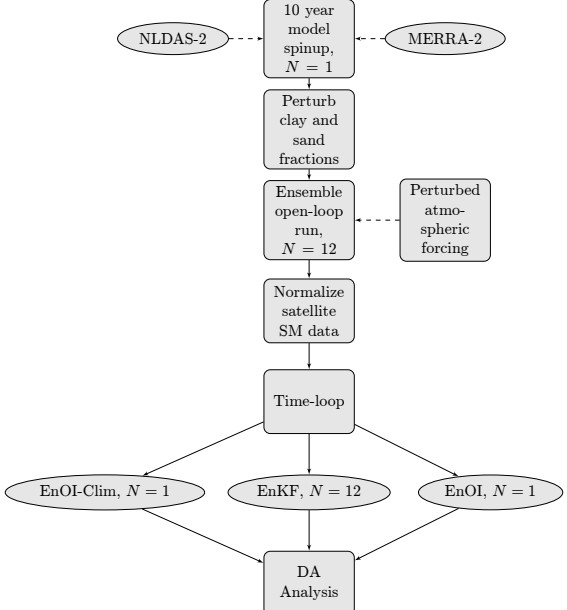

**Figure 1.** The experiments are setup using a 10 year spinup for the land surface initial conditions. From the initial state, we perform three different experiments: (i) the EnKF with evolving background error; (ii) the EnOI using the background error from the ensemble open-loop run, and (iii) the EnOI using a climatological background error as the ensemble spread.

The output SM is given for the eight top soil layers, which represent the first metre of soil. We create a surface zone soil moisture (sfzsm, layers 1 to 2) and a root-zone soil moisture (rzsm, layers 1 to 8) product, from weighting the different layers according to their depth. We use two sets of skill metrics to evaluate our experiments; one is the internal DA diagnostics, which check that the conditions for applying the different DA algorithms are met and also that the output is reasonable given the errors set in the system. In addition, we make sure that the biases in the system are removed and that the expected model and observation errors correspond to the actual errors. To diagnose the settings of the model and observation errors, we compute the standard deviation of the normalized innovations. This metric is given as:

$$\mathrm{Std}\left(\frac{\mathrm{O} - \mathrm{F}}{\mathbf{HBH}^T + \mathbf{R}}\right) \sim 1 \,, \tag{1}$$

where O is the observations and F is the model forecast, and the O-F is the observation-minus-forecast or innovations. If this metric is larger than 1, the actual errors are said to be underestimated, and overestimated if it is smaller than 1 [18,27]. In addition, we compute the lagged auto-correlation between the O-F pairs for each assimilation window, then take the average over the assimilation windows and the domain to create an auto-correlation function. The O-F sequence is assumed to behave as white noise in the Kalman Filter equations; if there are persistent biases, the auto-correlation function will differ from that of a white noise process. Hence, we can diagnose whether or not the system is bias free and close to behaving optimally [18].

The other metrics we use are the Pearson correlation coefficient (R), and the unbiased root mean square difference (ubrmsd), computed between the model outputs and the individual in situ stations. The 95% confidence interval (CI) for the correlation coefficient is computed using the Fisher

Z-transformation. The metrics are computed using all three hourly values as long as the in situ station has more than 150 measurements. The domain average metrics are computed by averaging over all the in situ stations, without any weighting depending of the clustering of the stations as in De Lannoy and Reichle [70]. This could mean that we are weighting densely sampled regions too much in our domain average metrics, and in the worse case lead to an overoptimistic CI.

**Table 2.** List of experiments performed. The open-loop run using NLDAS-2 forcing is given as eOL_NLDAS, while the open-loop run using MERRA-2 forcing is given as eOL_MERRA. The data assimilation runs are either the EnKF, EnOI or EnOI-Clim with the corresponding atmospheric forcing prescribed to the LSM. For the satellite skill experiments we also add M (MERRA-2) or N (NLDAS-2) to the experiment names depending on atmospheric forcing used. CPU cost is based on the total number of runs needed to do the analysis, for the EnKF this is $12 + 12$ (eOL + EnKF).

| Exp. | $N_{ens}$ | Param. Pert. | Rel. Obs. Error $\lambda$ | Satellite Data | Forcing | CPU Cost |
|---|---|---|---|---|---|---|
| eOL_MERRA2 | 12 | yes | - | - | MERRA-2 | 12 |
| EnKF_MERRA2 | 12 | yes | 0.3 | SMAP | MERRA-2 | 24 |
| EnOI_MERRA2 | 1 | yes | 0.3 | SMAP | MERRA-2 | 13 |
| EnOI_MERRA2-Clim | 1 | no | 0.3 | SMAP | MERRA-2 | 2 |
| eOL_NLDAS | 12 | yes | - | - | NLDAS-2 | 12 |
| EnKF_NLDAS | 12 | yes | 0.3 | SMAP | NLDAS-2 | 24 |
| EnOI_NLDAS | 1 | yes | 0.3 | SMAP | NLDAS-2 | 13 |
| EnOI_NLDAS-Clim | 1 | no | 0.3 | SMAP | NLDAS-2 | 2 |
| Satellite skill experiments | | | | | | |
| EnOI_ESA_M | 1 | yes | 0.3 | ESA CCI | MERRA-2 | 13 |
| EnOI_SMAP_M | 1 | yes | 0.3 | SMAP | MERRA-2 | 13 |
| EnOI_SMOS_M | 1 | yes | 0.3 | SMOS | MERRA-2 | 13 |
| EnOI_ESA_N | 1 | yes | 0.3 | ESA CCI | NLDAS-2 | 13 |
| EnOI_SMAP_N | 1 | yes | 0.3 | SMAP | NLDAS-2 | 13 |
| EnOI_SMOS_N | 1 | yes | 0.3 | SMOS | NLDAS-2 | 13 |

## 5. Data Assimilation Results and Discussion

### 5.1. Filter Performance EnKF vs. EnOI Using Data Assimilation Diagnostics

In this section, we compare the EnKF vs. EnOI using output information from the DA system, these skill metrics indicate how and if the DA system is performing according to its underlying assumptions. First, we compute the domain average filter performance, this is done separately for each of the assimilation windows; then, a final aggregation is applied to give the domain average for a given day. Figure 2a shows the mean O-F and O-A difference for the EnKF; plots for the EnOI and EnOI-Clim are not shown as the results are very similar to that of the EnKF. We notice that there is a slight wet bias of $0.01 \pm 0.01$ m$^3$m$^{-3}$ during the two summer seasons for the EnKF, this is also seen for the EnOI and EnOI-Clim (not shown). In general, the filters are moving the model forecast closer to the observations (the O-F difference is on average larger than the O-A difference). From Figure 2a, we interpret the system as close to bias free (long term mean close to zero), thus the CDF-matching has removed most of the systematic errors between the model and satellite data.

Figure 2b shows the standard deviation of the O-F and O-A differences; this illustrates the typical size of the O-F/A difference. The long-term means of the O-F and O-A spatial standard deviations are $0.0136/0.0105$, $0.0139/0.0102$ and $0.0139/0.011$ m$^3$m$^{-3}$ for the EnKF, EnOI and EnOI-Clim, respectively. The largest differences in standard deviations of O-F vs. O-A are seen for the EnOI. An explanation for this is that in the EnOI the ensemble spread does not change after the analysis, while, for the EnKF, the ensemble spread decreases after the analysis. The smaller spread will in turn result in a larger weight to the model forecast, hence the corrections towards the satellite observations will not be as large as in the EnOI.

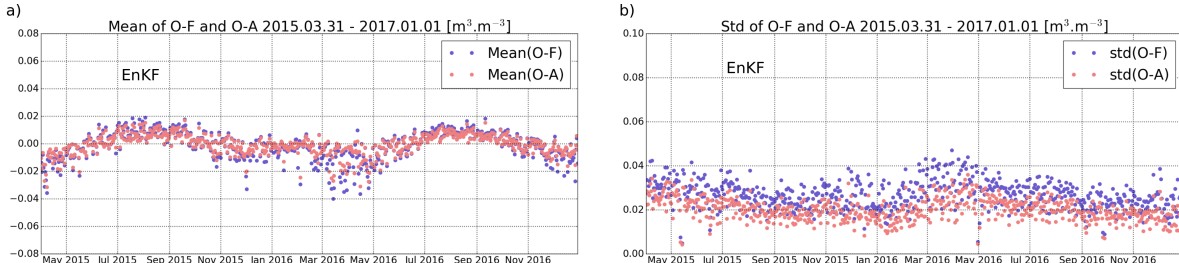

**Figure 2.** (**a**) domain average Observation-minus-Forecast (O-F) soil moisture ($m^3m^{-3}$) (blue) and Observation-minus-Analysis soil moisture ($m^3m^{-3}$) (red) residual for the EnKF; (**b**) domain standard deviation of O-F ($m^3m^{-3}$) (blue) and O-A soil moisture ($m^3m^{-3}$) (red) for the EnKF.

In Figure 3a–c, the instantaneous O-F residuals are given for the EnKF, EnOI and EnOI-Clim. The differences between the O-F residuals are small for the three filters, indicating that the model runs are quite similar, and that none of the filters introduces biases in their analysis. The observation layer increments (analysis-minus-forecast, not shown), show minor differences for the EnKF and the EnOI, which means that the background error-covariances in the two filters are close to identical. This suggests that the updates in the background error-covariance of the EnKF in the analysis step are minor (at least until 16 May 2015).

Figure 3d–f show the instantaneous increments for the top four layers; for the EnKF and the EnOI, the magnitude of the increments are close to those in the observation layer (plot not shown), while, for the EnOI-Clim, they are slightly smaller. By comparing Figure 3a–c with Figure 3d–f, we see that the three filters add/remove water to/from the top four layers when the observations are wetter/drier than the model forecast. For the EnKF and EnOI, an O-F residual of $\pm0.1\,m^3m^{-3}$ is translated into an increment of $\pm0.02\,m^3m^{-3}$ in the top four soil layers. The EnOI-Clim has the smallest increments; this is likely because the climatological background error underestimates the actual background error for this analysis step.

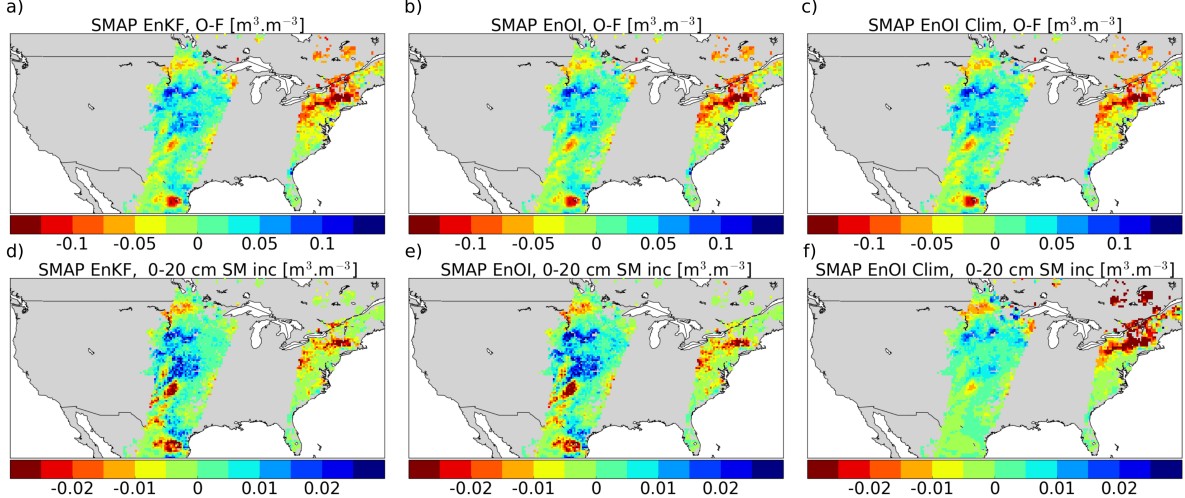

**Figure 3.** Instantaneous analysis output from 16 May 2015 at 1200 UTC for the EnKF, EnOI and EnOI-Clim; (**a–c**) observation-minus-Forecast (O-F) soil moisture residual ($m^3m^{-3}$) for the model observation layer; (**d–f**) soil moisture increments ($m^3m^{-3}$) for the top four soil layers 0–20 cm. Blue/red colours indicate positive/negative O-F residuals and increments.

A soil-layer vs. correlation plot (not shown) indicated that the correlation between the different soil layers decreased to zero at layer five, before it increased again. This was likely due to spurious correlations between the observation layer and the deeper layers. It was difficult to judge how physically consistent the updates in the lower layers were, compared to the update in the observation layer. For this reason, we only update the top four layers in the ISBA model, instead of updating the

top eight layers (representing the first metre of soil). We therefore found it better to let the model physics transfer the analysis information further down in the root-zone.

Spatial maps of the O-F standard deviations are shown in Figure 4a–c, the figures illustrate the typical spatial residual between the SMAP observations and the model forecast. The domain averages for the three filters are 0.0354 (EnKF), 0.0355 (EnOI) and 0.0354 m$^3$m$^{-3}$ (EnOI-Clim), which is close to what De Lannoy and Reichle [70] found for SMOS SM innovations. Typically, dry/wet regions (west/east) have small/large O-F differences, respectively. Large O-F differences are especially found in the Rocky and Appalachian mountain ranges. Soil moisture variability is expected to be higher in the mountainous regions, which might be a cause for the larger O-F differences seen here. The differences in the spatial patterns between Figure 4a,b are minor, indicating that the EnKF (flow dependent background error) and the EnOI (with the ensemble open-loop static background error) are very similar.

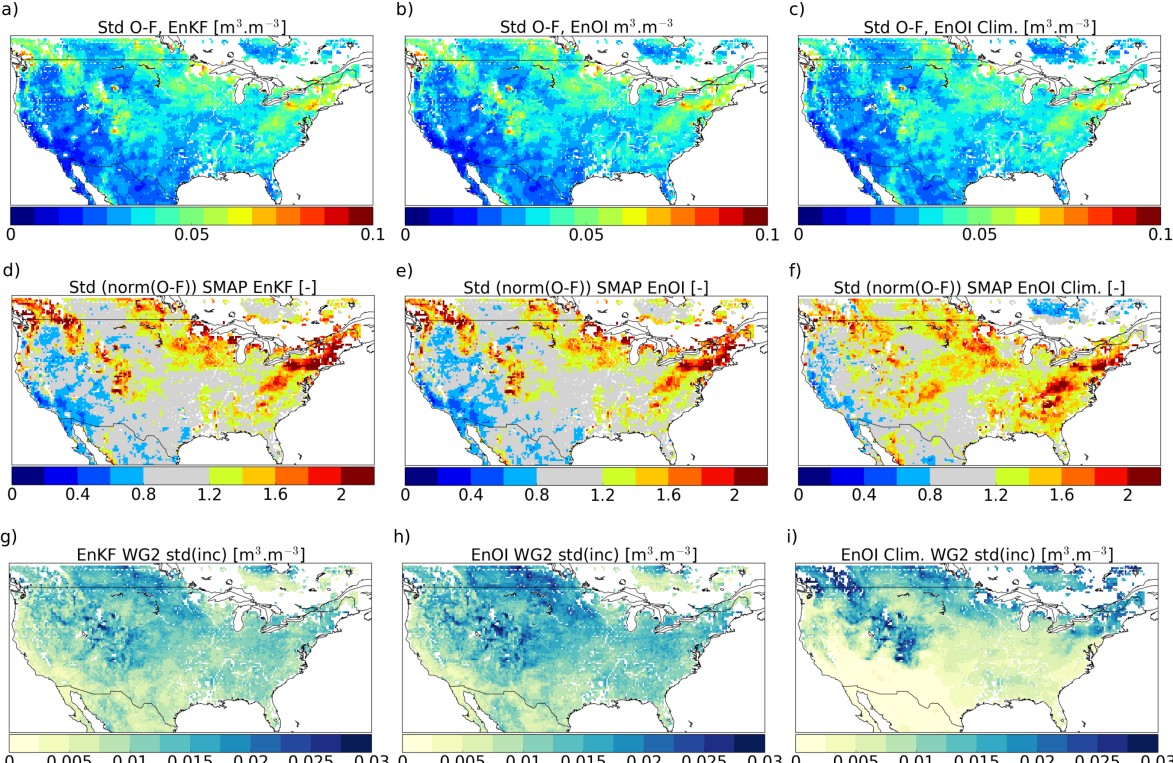

**Figure 4.** Assimilation diagnostics for the period from 31 March 2015 to 31 December 2016, for the EnKF, EnOI and EnOI-Clim. (**a–c**) standard deviation of the Observation-minus-Forecast (O-F) soil moisture residual (m$^3$m$^{-3}$), blue/red colours indicate low/high O-F residuals; (**d–f**) standard deviation of normalized innovations (dimensionless), blue/red colours indicate too large/small assumed errors; (**g–i**) standard deviation of the observation layer soil moisture increments (m$^3$m$^{-3}$), yellow/blue colours indicate low/high typical increments.

Figure 4d–f show spatial maps of the standard deviation of the normalized innovations—see Equation (1) for a detailed description. A value close to 1 (grey regions) means that the assumed errors are close to the actual errors for this grid-cell. Values smaller than 1 (blue regions) have assumed errors that are typically too large compared to the actual errors, while red regions (values larger than 1) have assumed errors that are typically smaller than the actual errors. Domain average values for the standard deviation of the normalized innovations are 1.18, 1.15 and 1.18 (dimensionless), for the EnKF, EnOI and EnOI-Clim, respectively. The spatial patterns of the normalized innovations are close to identical for the EnKF and EnOI; the EnOI-Clim has a domain average very close to that of the EnOI and EnKF, even though the spatial pattern differs. The blue regions in the west (too large assumed

errors) are not as prominent in the EnOI-Clim filter. This is likely because of a low ensemble spread in the dry regions for the climatologically sampled background error in the EnOI-Clim. Regions where the assumed errors are too small (red) cover larger parts of the domain for the EnOI-Clim compared to the EnKF and EnOI. A likely reason for this is that the computation of the climatology averages out precipitation events, leading to a small spread in the ensemble.

Comparing Figure 4a–c to Figure 4d–f, we notice that regions with large O-F residuals tend to be regions where the model and observation errors are underestimated, i.e., too low spread or too low observation error, or both, while regions with low O-F residual have model and observation errors that are too large (blue regions). Again, we see that there is not much difference between an evolving background error in the EnKF and a static ensemble background error in the EnOI.

Figure 4g–i show the standard deviations of the observation layer increments. The domain average of a typical observation layer increment for the EnKF, EnOI and EnOI-Clim are 0.008, 0.009 and 0.007 $\mathrm{m^3m^{-3}}$. The EnKF and EnOI spatial patterns are very similar, with a lower magnitude for the EnKF than the EnOI, which is likely due to the analysis update of the background error. Dry regions in the south and southwest have smaller increments than the vegetated wet regions in the mountains and to the east. Typical increments for the EnOI-Clim are small in the south and southwest, most likely because the climatology is not able to create a large enough ensemble spread, while the mountainous regions have larger increments because of a greater variability in SM, which is likely increasing the ensemble spread.

The final assessment of the EnKF, EnOI and EnOI-Clim filter performance is done by looking at the lagged auto-correlation between pairs of O-F residuals. From DA theory, the assumption is that the innovations should be white noise for the filter to perform close to "optimal". Lag is given in days and averaged over the different analysis times and over the spatial domain. The plot (not shown) of the lagged autocorrelation yields similar results for all three filters. The auto-correlation drops towards zero (values between 0.05 and 0.2) but does not follow the same pattern as white noise. This could be an indication of a remaining bias between the model and the observations. However, it is small and in alignment with what other studies have found for SMAP DA [18].

Overall, the DA statistics for the EnKF and EnOI show that, from the perspective of the DA diagnostics, they perform similarly. For the EnOI-Clim, further study is needed to understand how best to capture the errors of the day in a climatologically sampled background error-covariance.

*5.2. EnKF vs. EnOI; Comparison with In Situ SM Data*

5.2.1. Correlation Skill

While internal DA diagnostics are useful for evaluating the land DA system's consistency with regard to DA assumptions, it does not convey any information about the system performance relative to independent data. For such an evaluation, we use the SCAN and USCRN in situ networks. Figure 5a shows the Pearson R correlation coefficient between the ensemble open-loop run using MERRA-2 forcing (eOL_MERRA2) and the SCAN (triangles) and the USCRN (circles) stations. The eOL_MERRA2 run alone clearly has good skill in the southeast, while, for example, in the mountainous regions (Rockies), it shows lower correlation values. The domain average R between the eOL_MERRA2 and SCAN is 0.62, while it is 0.66 between the eOL_MERRA2 and USCRN; the domain average surface zone statistics are summarized in Table 3. For the surface zone correlation, both the EnKF (0.67/0.7) and the EnOI (0.66/0.7) are found to improve the correlation with the SCAN and USCRN in situ stations at the 95% significance level, respectively.

A difference plot of the EnKF_MERRA2-eOL_MERRA2 (ΔR) correlations is given in Figure 5b. Here we note that regions in Figure 5a showing lower R-values, tend to have a positive ΔR, while for regions in the southeast, east and west (California), there is little change in ΔR. The largest positive impact from the land DA is seen in the Midwest. The high number of in situ stations with improved

correlations make us confident that the large scale error settings (for model and observation errors) in the EnKF are reasonable.

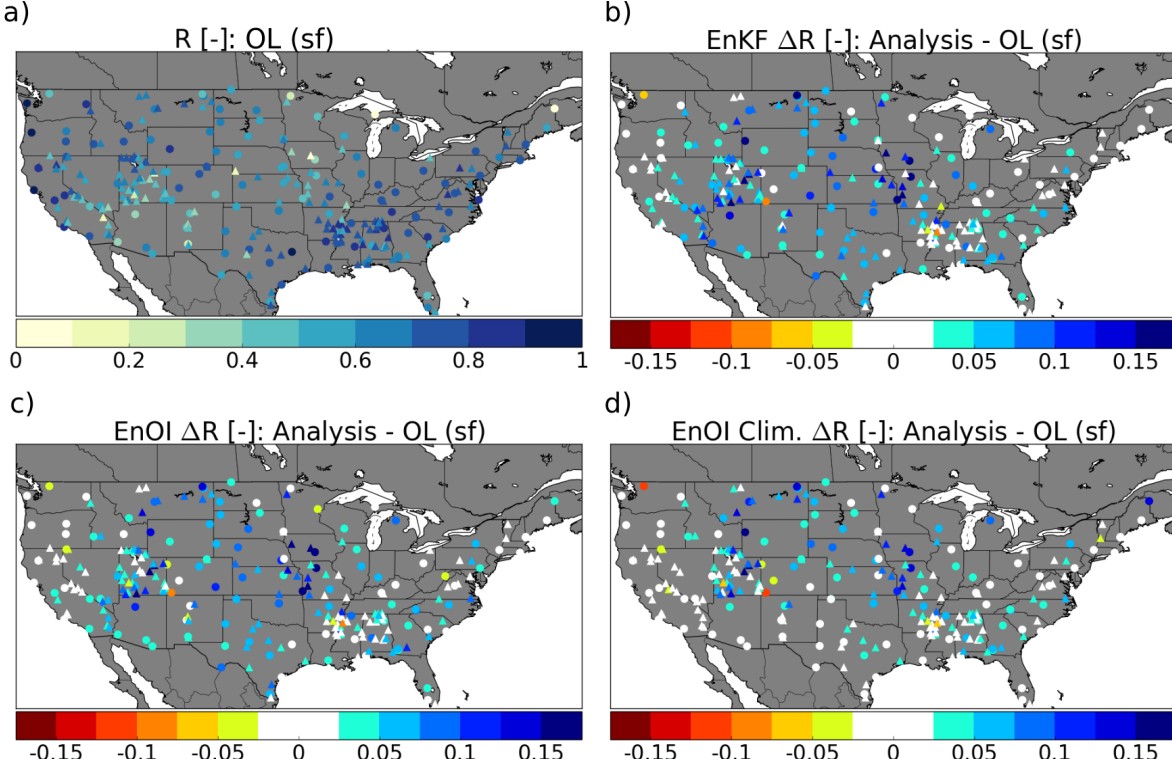

**Figure 5.** (**a**) Pearson R correlation coefficient between the ensemble open-loop (eOL) surface zone soil moisture and the SCAN (triangles) and USCRN (circles) stations. Yellow/blue indicate low/high correlation; (**b**) ΔR (EnKF correlation minus eOL correlation), blue/red colours indicate positive/negative impact (226 of 259 stations improved); (**c**) ΔR EnOI (190 of 259 stations improved); (**d**) ΔR EnOI-Clim (183 of 259 stations improved).

For the EnOI_MERRA2, in Figure 5c, much of the spatial pattern is the same as for the EnKF_MERRA2 in Figure 5b; however, more stations show a negative impact. As for the EnKF_MERRA2, the largest positive impact is seen in the Midwest. We emphasize that, in the difference plots, we have not taken into account if the individual station improvements are statistically significant or not in the colour coding; we only compute the difference in correlation and count the number of positive vs. negative values. The significance of the improvements is taken into account when we calculate the domain average statistics found in Table 3.

The EnOI_MERRA2-Clim in Figure 5d shows a band going from the northwest to the southeast where there is little change in R; and this region is also the region with the smallest standard deviation of the increments in Figure 4. For the EnOI_MERRA2-Clim, there are more stations with negative values for the ΔR than for the EnKF and EnOI.

The root-zone eOL_MERRA2 SM correlations with the in situ networks are given in Figure 6a; and, again, we notice high correlations in the southeast and east. Lower correlations are found in the Midwest and in the mountainous region (Rockies). Computing the domain average for the eOL_MERRA2 yields an R of 0.63 and 0.68 for the SCAN and USCRN stations, domain average statistics for the root-zone SM can be found in Table 3.

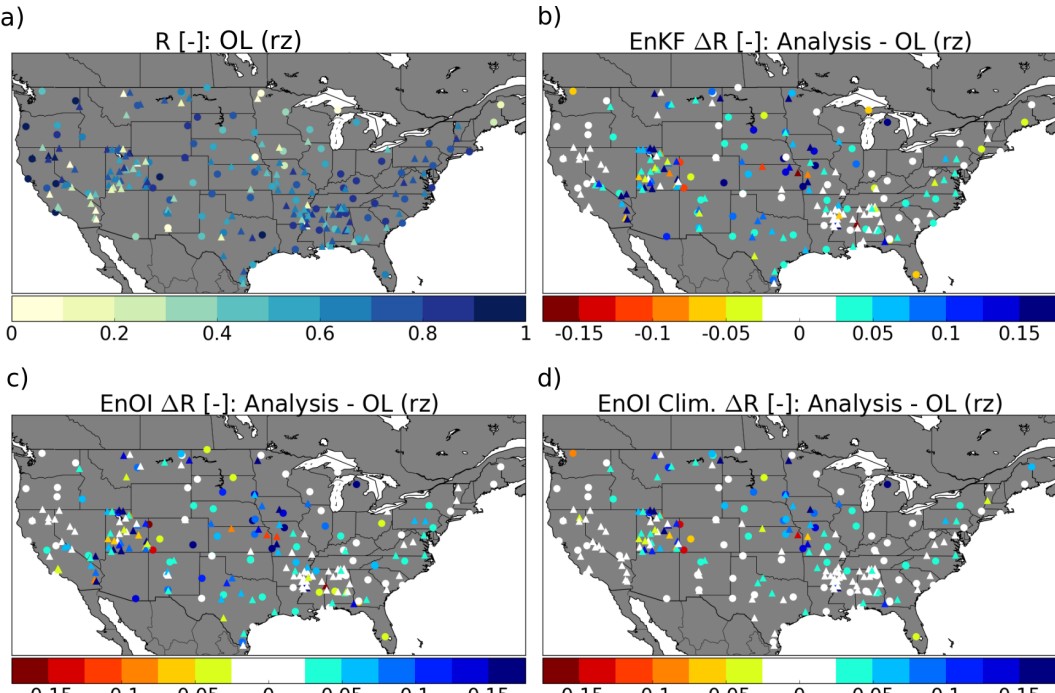

**Figure 6.** (**a**) Pearson R correlation coefficient between the ensemble open-loop (eOL) root-zone soil moisture and the SCAN (triangles) and USCRN (circles) stations. Yellow/blue indicate low/high correlation. (**b**) ΔR (EnKF correlation minus eOL correlation), blue/red colours indicate positive/negative impact (153 of 223 stations improved). (**c**) ΔR EnOI (135 of 223 stations improved), (**d**) ΔR EnOI-Clim (145 of 223 stations improved).

The EnKF_MERRA2 ΔR in Figure 6b does not show as much improvement as for the surface zone. There are more stations with a negative impact and the size of the negative impact is larger than for the surface zone. For the root-zone, there are fewer stations with enough data for the intercomparison, thus the total number of stations is smaller. Although the stations with positive impact from the land DA are more scattered than for the surface zone, we find that most of the stations with the largest improvements in the EnKF_MERRA2 analysis are seen in the Midwest. The largest positive impacts from the EnOI_MERRA2 DA analysis are also in the Midwest (see Figure 6c). For the EnOI_MERRA2-Clim in Figure 6d, the impact of the DA analysis is lower (more stations with white colour coding, showing little positive or negative impact), and again there is a band from west–southeast with little change, and this is most likely due to small increments in the analysis. Although the changes overall in correlations are smaller than for the EnOI_MERRA2, the total number of stations with improved correlations are larger for the EnOI_MERRA2-Clim.

The summary statistics of the eOL, EnKF, EnOI and EnOI-Clim assimilation of SMAP data are given in Figure 7 and Table 3. In Figure 7a, the EnKF and EnOI driven by MERRA-2 forcing show a significant improvement in the surface zone (non-overlapping confidence intervals) for both the SCAN and USCRN networks, when compared to the eOL_MERRA2. In the same figure, only for EnKF, EnOI and eOL driven by NLDAS forcing, we see that there is an improvement, but it is not statistically significant (i.e., overlapping confidence intervals). One reason for this might be the already high correlation between the eOL_NLDAS and the in situ stations. The NLDAS precipitation correction filters out random errors in the forcing data, which leaves little room for improvement in the land DA analysis. We also note that the EnKF using MERRA-2 forcing has almost the same skill as the eOL_NDLAS. This surface zone improvement in the EnKF_MERRA2 analysis towards an eOL_NLDAS run driven by observation corrected precipitation shows that assimilation of SMAP-L2 data could be a promising way to improve the land surface state in regions of the world where high-quality meteorological data are unavailable.

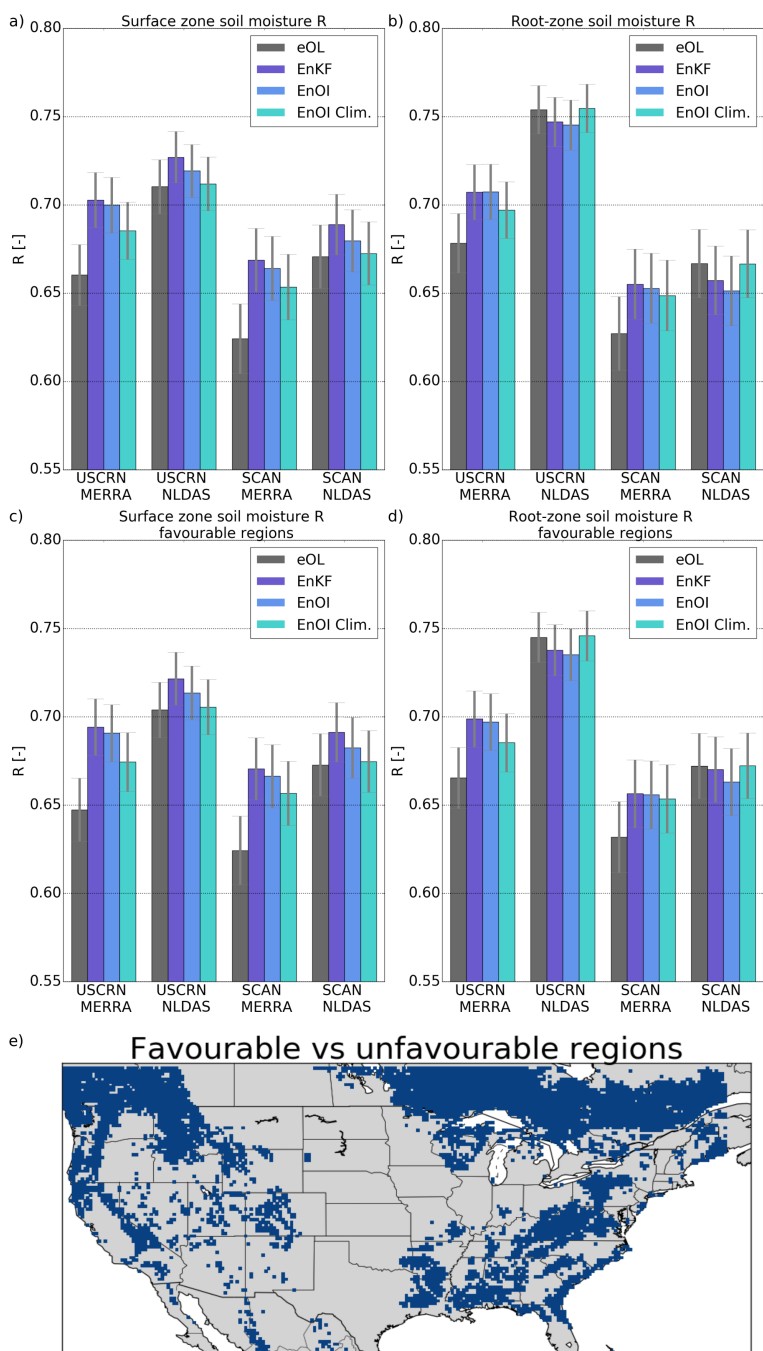

**Figure 7.** Summary statistics for the Pearson R correlation coefficient between the model and in situ data, a 95% confidence interval for the domain averaged correlation values are given as grey error bars. (**a**) domain average surface zone R, eOL (black), EnKF (dark blue), EnOI (light blue) and EnOI-Clim (turquoise); (**b**) domain average root-zone R; (**c**) domain average R "favourable" regions surface zone; (**d**) domain average R "favourable" regions root-zone; (**e**) favourable (grey land pixels) vs. unfavourable (blue land pixels) regions for soil moisture DA. See text for description of "favourable" vs. "unfavourable" regions.

**Table 3.** Surface and root-zone Pearson R correlation coefficient and unbiased root mean square difference (ubrmsd) with SCAN and USCRN in situ networks. Confidence intervals (CI) are given in half values of the 95% interval for the individual experiments. The experiments with highest skill metrics are highlighted in bold.

| Exp. | SCAN [a,b] | | | USCRN [c,d] | | |
|---|---|---|---|---|---|---|
| | R | CI | Ubrmsd | R | CI | Ubrmsd |
| sfzsm $(0-5)$ cm, $(m^3 m^{-3})$ | | | | | | |
| eOL_MERRA | 0.62 | ±0.020 | 0.057 | 0.66 | ±0.017 | 0.052 |
| EnKF_MERRA2 | **0.67** | **±0.018** | **0.053** | **0.70** | **±0.016** | **0.049** |
| EnOI_MERRA2 | 0.66 | ±0.018 | 0.054 | 0.70 | ±0.016 | 0.049 |
| EnOI_MERRA2-Clim | 0.65 | ±0.019 | 0.054 | 0.68 | ±0.016 | 0.050 |
| eOL_NLDAS | 0.67 | ±0.018 | 0.053 | 0.71 | ±0.015 | 0.048 |
| EnKF_NLDAS | **0.69** | **±0.017** | **0.052** | **0.73** | **±0.015** | **0.047** |
| EnOI_NLDAS | 0.68 | ±0.018 | 0.052 | 0.72 | ±0.015 | 0.048 |
| EnOI_NLDAS Clim | 0.67 | ±0.018 | 0.053 | 0.71 | ±0.015 | 0.048 |
| rzsm $(0-100)$ cm, $(m^3 m^{-3})$ | | | | | | |
| eOL_MERRA | 0.63 | ±0.021 | 0.041 | 0.68 | ±0.017 | 0.042 |
| EnKF_MERRA2 | **0.66** | **±0.020** | **0.040** | **0.71** | **±0.016** | **0.040** |
| EnOI_MERRA2 | 0.65 | ±0.020 | 0.040 | 0.71 | ±0.016 | 0.040 |
| EnOI_MERRA2-Clim | 0.65 | ±0.020 | 0.040 | 0.70 | ±0.016 | 0.041 |
| eOL_NLDAS | **0.67** | **±0.019** | **0.041** | **0.75** | **±0.014** | **0.041** |
| EnKF_NLDAS | 0.66 | ±0.019 | 0.041 | 0.75 | ±0.014 | 0.041 |
| EnOI_NLDAS | 0.65 | ±0.020 | 0.041 | 0.75 | ±0.014 | 0.041 |
| EnOI_NLDAS Clim | 0.67 | ±0.019 | 0.041 | 0.75 | ±0.014 | 0.041 |

Number of stations: [a] $N_{sfzsm} = 151$, [b] $N_{rzsm} = 138$, [c] $N_{sfzsm} = 108$, [d] $N_{rzsm} = 85$.

Figure 7b and Table 3 show the domain average root-zone correlation between the EnKF, EnOI, EnOI-Clim and the eOL with the in situ stations. The EnKF and EnOI driven by MERRA-2 forcing have equal skill (R = 0.71 for both DA algorithms) for the comparison with the USCRN network, and the improvement relative to the eOL (R = 0.68) is close to being statistically significant at the 95% level. The EnOI-Clim also shows an improvement when compared to the USCRN (R = 0.7); however, the improvement is not significant at the 95% level. For SCAN using MERRA-2 forcing, all of the filters show improvements compared to the eOL_MERRA2, but none are statistically significant. Using the NLDAS forcing, both the USCRN and SCAN comparisons show that the filters have a lower skill in the root-zone than the eOL_NLDAS; however, the difference is not statistically significant. Here, the EnOI_NLDAS Clim has the best skill, most likely because of the small corrections done by the filter in some regions of the domain (see discussion in Section 5.1).

Figure 7c,d are the same as Figure 7a,b, except they show the statistics for the regions favourable for soil moisture DA. These are regions where the water fraction in the grid-cell is below 5%, (i.e., excluding lakes), there is no permanent snow or ice, and there is little dense vegetation, i.e., forest, and/or no strong topography in the grid-cell. The only statistically significant improvement in root-zone SM is seen for the EnKF over USCRN stations located in regions favourable for soil moisture DA. Here, the correlation value increases from 0.67 (eOL_MERRA2) to 0.7 (EnKF). The overall model performance (eOL) is however found to be lower when computed for favourable areas than for the whole domain. A likely reason for this is that the stations in the southeast with high initial correlations are left out of the overall statistics, since they are in a region unfavourable for soil moisture DA. It is expected that the DA analysis will have very little impact over this region, hence it is not included in the overall statistics. The favourable vs. unfavourable regions are in this way not reflecting regions with high initial eOL skill, but regions where we expect that the DA can have a positive impact, regardless of the initial skill of the eOL.

Overall, there were small differences between the EnKF and EnOI filters, while the EnOI-Clim had the lowest correlation skill (Table 3). The small differences between the EnKF and EnOI can be explained by two things: (i) the relatively long time-window between observations in the same

grid-cell allows the spread in the EnKF to approach that of the EnOI, and (ii) the non-chaotic behaviour of the system suppresses the importance of the initial conditions in the forward model integrations. Points (i) and (ii) imply that it is limited how different a forward integration of an ensemble (EnKF) can be from that of a single model trajectory (EnOI). Skill improvements were smaller when utilizing observation corrected forcing (NLDAS); however, we did see that, when utilizing a model derived forcing (MERRA-2) and land DA, we were able to get comparable skill between the land DA analysis and the ensemble open-loop run with observation corrected forcing. We still saw these improvements using the EnOI highlighting the potential use of this computationally cheaper method for estimating soil moisture.

### 5.2.2. Unbiased Root Mean Square Difference

In Figure 8a, we have plotted the surface zone unbiased root mean square difference (ubrmsd), between the eOL (using MERRA-2 forcing) and the SCAN (triangles) and USCRN (circles) networks. Low/high ubrmsd values are depicted with yellow/blue values, respectively. The lowest ubrmsd values are found in regions where the SM variability is naturally limited (dry regions to the west), while higher values are found where the SM variability is high (the wetter east). The Δubrmsd for the EnKF_MERRA2 minus the eOL_MERRA2 metric is plotted in Figure 8b, blue regions indicate that the EnKF_MERRA2 has improved the ubrmsd skill and the opposite for red regions. Both the EnKF_MERRA2 and EnOI_MERRA2 show the largest increase in ubrmsd skill in the Midwest, while the west coast and mountainous region have lower to no increase in ubrmsd skill. For the EnOI_MERRA2-Clim, little ubrmsd change is seen in the west (along the Pacific coast) and in the south (Arizona, New Mexico and Texas), which is likely linked to the smaller analysis increments found in these regions—see Section 5.1.

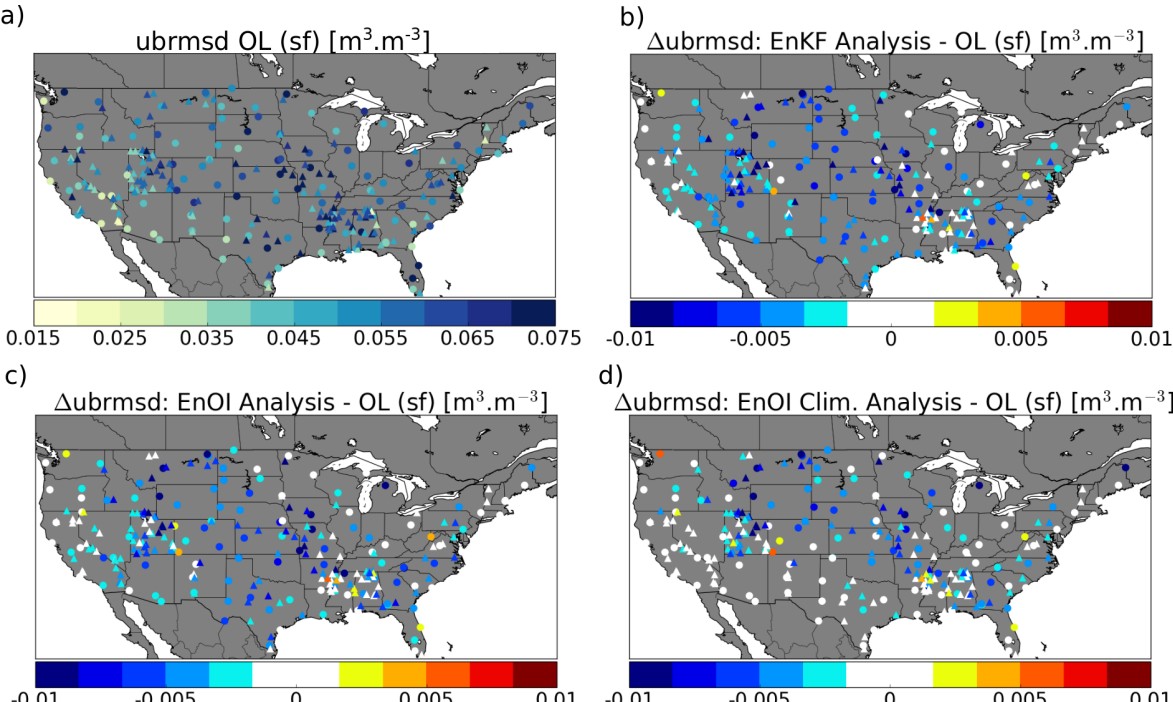

**Figure 8.** (**a**) unbiased root mean square difference (ubrmsd, $m^3 m^{-3}$) between the ensemble open-loop (eOL) surface zone soil moisture and the SCAN (triangles) and USCRN (circles) stations. Yellow/blue colours indicate low/high ubrmsd; (**b**) Δubrmsd (EnKF minus eOL); blue/red colours indicate positive/negative impact of the analysis (229 of 259 stations improved); (**c**) Δubrmsd (EnOI minus eOL) (201 of 259 stations improved); (**d**) Δubrmsd (EnOI-Clim minus eOL) (197 of 259 stations improved).

The root-zone eOL_MERRA2 ubrmsd for the SCAN (triangles) and USCRN (circles) networks are plotted in Figure 9a, low/high values are depicted in yellow/blue colours, respectively. The natural variability of root-zone SM is lower than that of the surface, hence more stations have lower values in this figure than in Figure 8a. Figure 9b is the root-zone Δubrmsd between the EnKF_MERRA2 and the eOL_MERRA2. Improvements (blue colours) are mostly found in the Midwest, while most negative EnKF_MERRA2 analysis impacts are found in the southeast, east and in the mountain ranges to the west. Figure 9c, shows the Δubrmsd for the EnOI_MERRA2, as for the EnKF_MERRA2 most improvements are found in the Midwest. As seen for the surface zone, the domain average absolute value is smaller for the EnOI_MERRA2-Clim than for the EnKF_MERRA2 and the EnOI_MERRA2. A likely reason for the EnOI_MERRA2-Clim to have more stations with positive improvements is that the analysis increments are smaller for this filter.

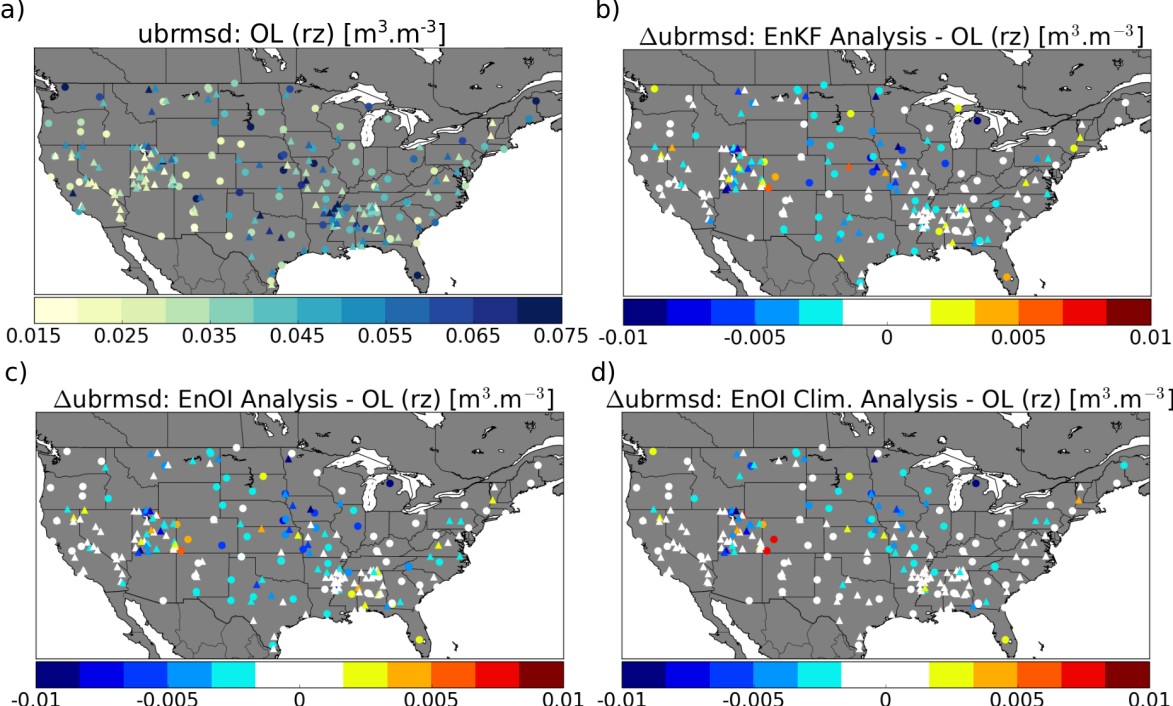

**Figure 9.** (**a**) unbiased root mean square difference (ubrmsd, $m^3 m^{-3}$) between the ensemble open-loop (eOL) root-zone soil moisture and the SCAN (triangles) and USCRN (circles) stations. Yellow/blue colours indicate low/high ubrmsd; (**b**) Δubrmsd (EnKF minus eOL), blue/red colours indicate positive/negative impact of the analysis (148 of 223 stations improved); (**c**) Δubrmsd (EnOI minus eOL) (127 of 223 stations improved); (**d**) Δubrmsd (EnOI-Clim minus eOL) (130 of 223 stations improved).

Figure 10a–d present a summary of the ubrmsd statistics for all stations and stations located in regions favourable for soil moisture DA. For the surface zone, in Figure 10a, the largest improvements are seen for the EnKF_MERRA2 ($0.053/0.049\,m^3 m^3$), while the EnOI_MERRA2 ($0.054/0.049\,m^3 m^3$) and EnOI_MERRA2-Clim ($0.054/0.05\,m^3 m^3$) also improve, relative to the eOL_MERRA2 ($0.057/0.052\,m^3 m^3$), for SCAN and USCRN, respectively.

The EnKF_MERRA2 has a spatial average for the surface Δubrmsd of $-0.0036\,m^3 m^3$; this is larger in magnitude than the EnOI_MERRA2 which has a value of $-0.0031\,m^3 m^3$ and the EnOI_MERRA2-Clim with $-0.0024\,m^3 m^3$. These values are close to what De Lannoy and Reichle [12] report in their study for the EnKF assimilating SMOS SM retrievals. Little change is seen in surface ubrmsd for the EnKF_NLDAS, EnOI_NLDAS and EnOI_NLDAS-Clim when compared to the eOL_NLDAS. However, as for the correlation, we note that the ubrmsd skill of the EnKF_MERRA2 is close to that of the eOL_NLDAS.

A root-zone ubrmsd target value of $0.04\,\mathrm{m^3m^3}$ is set in the literature for SMAP-L4 products over favourable areas [38,49]. It is useful to compare our analysis results with that of the SMAP-L4 requirements. From the summary statistics in Figure 10d, we see that the SMAP-L4 requirements are met for all three filters over the USCRN stations using MERRA2 forcing. For the model runs using NLDAS forcing, the requirement is already met for the open-loop run over the USCRN stations. The domain average of the $\Delta$ubrmsd for all stations is smaller than for the surface zone, $-0.001\,\mathrm{m^3m^3}$ (EnKF_MERRA2), $-0.001\,\mathrm{m^3m^3}$ (EnOI_MERRA2) and $-0.0007\,\mathrm{m^3m^3}$ (EnOI_MERRA-Clim).

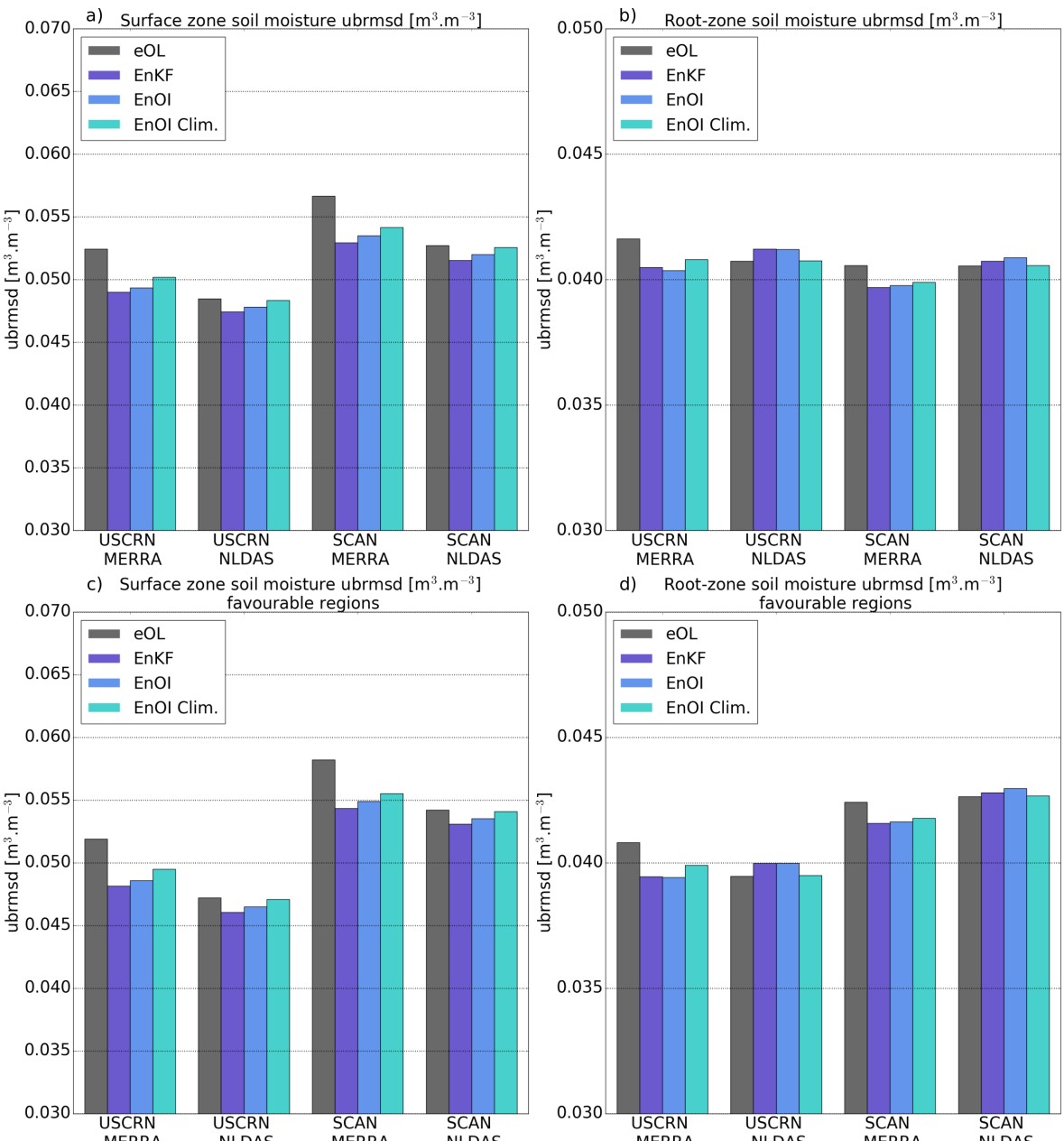

**Figure 10.** Summary statistics for the unbiased root mean square difference (ubrmsd, $\mathrm{m^3m^{-3}}$) between the model and in situ data, computed for all stations (**a**,**b**) and favourable regions (**c**,**d**). (**a**) domain average surface zone ubrmsd, eOL (black), EnKF (dark blue), EnOI (light blue) and EnOI-Clim (turquoise); (**b**) domain average root-zone ubrmsd.

Interpreting the improvements in ubrmsd is difficult because, for dry regions, the ubrmsd is naturally low, since the variability of SM for dry regions is low [17,49]. Looking at the spatial distribution of the surface zone Δubrmsd, in Figure 8b,c, most of the stations show negative values (positive analysis impact). This indicates that the improvements are due to the EnKF_MERRA2 and the EnOI_MERRA2 analysis, and not just a statistical artifact. We cannot say the same for the root-zone, as the spatial patterns here are more a mix between an increase and a decrease in skill. In addition, as for dry regions, root-zone SM has a naturally low variability making it difficult to interpret the results.

*5.3. Satellite Skill: Comparison between ESA CCI, SMAP and SMOS Using the EnOI*

In this section, we evaluate the results of the assimilation of the ESA CCI, SMAP and SMOS products. As in the intercomparison of the EnKF, EnOI and EnOI-Clim filters, we evaluate the results in this section against in situ stations. We apply the EnOI with the eOL ensemble as the background error-covariance, since it was found to have comparable skill to the EnKF, while being computationally cheaper—see Sections 5.1 and 5.2. In the assimilation of the three datasets, we use the MERRA-2 forcing, and the ESA CCI, SMAP and SMOS time-series that are all normalized to the eOL run using CDF-matching prior to the assimilation. The assimilation of the three different satellite products and the corresponding evaluation is done for the 31.03.2015 to 31.12.2016 time-period.

Figure 11a–b and Table 4 show the summary of the domain average surface and root-zone correlation between the eOL and in situ, EnOI_ESA and in situ, EnOI_SMAP and in situ and EnOI_SMOS and in situ. The average metrics are only computed for stations covered by the ESA CCI product and in regions favourable for soil moisture DA, hence the domain mean differs from the one shown in Figure 7. The surface zone correlation in Figure 11a shows that the EnOI_MERRA2 assimilation of SMAP data improves the correlation from 0.63 and 0.64 (eOL) to 0.68 and 0.7 at the 95% significance level for the SCAN and USCRN networks, respectively. The assimilation of SMAP is also seen to have the largest positive impact on the surface ubrmsd skill when comparing the three L2 datasets (Table 4). For the EnOI_NLDAS analysis, only assimilation of SMAP data slightly improves the correlation when compared to the eOL run. The SMOS assimilation stays neutral, while the assimilation of the ESA CCI product has a decrease in skill compared to the eOL (Table 4).

**Table 4.** Satellite skill metrics at SCAN and USCRN networks using MERRA-2 atmospheric forcing. The Pearson R correlation coefficient with half values of the 95% confidence interval (CI) for the individual experiments. The experiments with highest skill metrics are highlighted in bold.

| Exp. | SCAN [a,b] | | | USCRN [c,d] | | |
|---|---|---|---|---|---|---|
| | R | CI | Ubrmsd | R | CI | Ubrmsd |
| sfzsm $(0-5)$ cm, $(m^3 m^{-3})$ | | | | | | |
| eOL_MERRA | 0.63 | ±0.019 | 0.061 | 0.64 | ±0.018 | 0.054 |
| EnOI_SMAP | **0.68** | **±0.017** | **0.057** | **0.70** | **±0.016** | **0.050** |
| EnOI_SMOS | 0.66 | ±0.018 | 0.058 | 0.68 | ±0.017 | 0.051 |
| EnOI_ESA | 0.65 | ±0.018 | 0.059 | 0.65 | ±0.018 | 0.053 |
| rzsm $(0-100)$ cm, $(m^3 m^{-3})$ | | | | | | |
| eOL_MERRA | 0.64 | ±0.020 | 0.043 | 0.63 | ±0.019 | 0.042 |
| EnOI_SMAP | **0.67** | **±0.019** | **0.042** | **0.68** | **±0.017** | **0.041** |
| EnOI_SMOS | 0.66 | ±0.019 | 0.043 | 0.66 | ±0.018 | 0.041 |
| EnOI_ESA | 0.67 | ±0.019 | 0.042 | 0.64 | ±0.018 | 0.042 |

Number of stations: [a] $N_{sfzsm} = 86$, [b] $N_{rzsm} = 86$, [c] $N_{sfzsm} = 47$, [d] $N_{rzsm} = 41$.

Summary statistics for the the root-zone correlations with in situ data are found in Table 4 and in Figure 11b. The EnOI_MERRA2 assimilation of SMAP shows a statistically significant improvement for the USCRN stations (from 0.63 to 0.68 at a 95% level), and an improvement from 0.64 to 0.67 for the SCAN stations. Both the SMOS and the ESA CCI assimilation using EnOI_MERRA2 are also found to increase the root-zone skill over the eOL run. A slight decrease in skill is seen for the EnOI_NLDAS

assimilation of SMAP and SMOS; however, the confidence intervals are still overlapping. As seen in the surface zone, we also see a decrease in skill using the ESA CCI in the EnOI_NLDAS assimilation. Spatial maps of ΔR (not shown) indicate that the reason for the decrease in skill is not due to individual outliers, but an overall decrease in skill over several stations. We emphasize that the overall skill is computed for the same stations for the ESA CCI, SMAP and SMOS analysis, hence individual station characteristics should not have an impact on the difference in overall statistics between the three datasets. The most likely reason for the SMAP-L2 and SMOS-L2 analysis to yield better results than the ESA CCI analysis is that they are more sensitive to soil moisture (L-band), while most of the observations utilized in the ESA CCI product were C-band, for both the active and passive data. There is reason to expect that, when the ESA CCI product integrates more SMAP and SMOS data, the skill of the ESA CCI assimilation will increase. The assimilation of SMAP and SMOS showed slightly better performance for SMAP than for SMOS. These two satellites both operate in the L-band and use the same radiative transfer equation ($\tau - \omega$ model). Al-Yaari et al. [74] compared SMAP and SMOS soil moisture data and found that SMAP had better skill. Our results agree with this skill difference between SMAP and SMOS as seen in [74], only here it is not seen by direct intercomparison of the retrieved soil moisture products, but via the SMAP land DA analysis performing better than the SMOS land DA analysis. They point at the difference in ancillary data (land cover dataset), model parameterizations, and RFI mitigation and detection to be some of the reasons for the difference in skill. In addition, the reported instrument error (for Level-1 data) is lower for SMAP ($\sim$1 K) compared to SMOS ($\sim$3–3.5 K) [75].

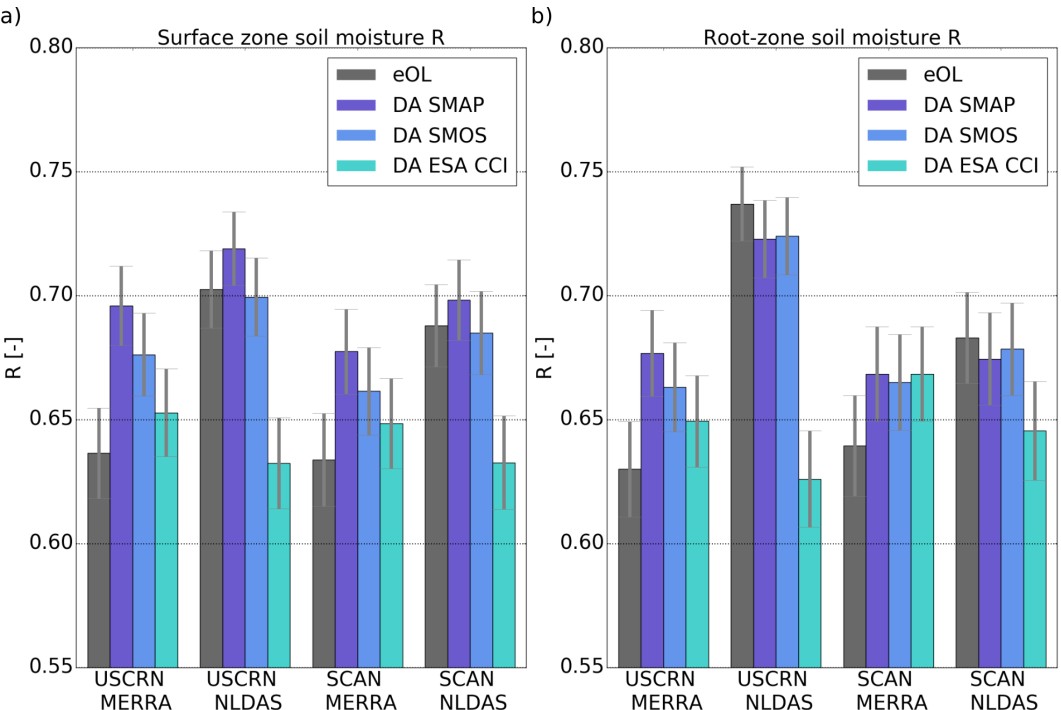

**Figure 11.** Summary statistics for the assimilation of SMAP (dark blue), SMOS (light blue) and ESA CCI (turquoise), surface and root-zone Pearson R correlation coefficient. Statistics are only computed for stations covered by the ESA CCI product; this is to ensure a fair intercomparison. (**a**) domain average surface zone R, eOL (black), SMAP (dark blue), SMOS (light blue) and ESA CCI (turquoise); (**b**) domain average root-zone R.

## 6. Conclusions

In this study, we first compared three different data assimilation (DA) methods, the EnKF, the EnOI and the EnOI-Clim by assimilating the SMAP-L2 soil moisture (SM) product over the contiguous United States (CONUS) using MERRA-2 meteorological forcing. The correlation and ubrmsd skill

of the resulting DA analysis was evaluated against independent in situ data from the Soil Climate Analysis Network (SCAN) and the U.S. Climate reference network (USCRN). It was shown that the EnKF had the highest correlation skill (average of the SCAN and USCRN networks) when compared to in situ stations (0.685 surface and 0.681 root-zone), while the correlation skill of the EnOI was close to that of the EnKF (0.68 surface and 0.68 root-zone). The EnKF ubrmsd skill (average of the SCAN and USCRN networks) was $0.051 \, \text{m}^3\text{m}^3$ (surface) and $0.04 \, \text{m}^3\text{m}^3$ (root-zone), again, the EnOI skill was close with ubrmsd of $0.0515 \, \text{m}^3\text{m}^3$ (surface) and $0.04 \, \text{m}^3\text{m}^3$ (root-zone). Given that the EnOI is computationally cheaper than the EnKF and it still offers much of the same benefits, it could be a promising alternative to the EnKF. The downside is that it does not use a flow dependent ensemble spread as a measure of the analysis uncertainty. This might not be that important though, as the assimilation window is so long that the ensemble spread of the EnKF is close to that of the ensemble open-loop (eOL) spread, making the differences in background error between the two filters minor. This can be explained by the nature of the soil moisture dynamics, which is stable with respect to initial conditions [76]. Thus, the findings may be extrapolated to other stable dynamical systems. The EnOI-Clim performed poorest, especially for dry regions, where the climatological spread was found to be too low. However, there should be potential to improve the EnOI-Clim results by, for example, looking more into sampling methods of the background error.

The impact of the atmospheric forcing on the analysis results was evaluated by assimilating SMAP data using the EnKF and the EnOI with MERRA-2 (model derived) and NLDAS (observation corrected) atmospheric forcing as model input. The largest improvements from an eOL run in correlation and ubrsmd were seen when we used the MERRA-2 atmospheric forcing data. This dataset had no precipitation correction, yet the initial skill was high when comparing the eOL forced by MERRA-2 data to in situ stations. The EnKF and the EnOI assimilation of SMAP showed statistically significant improvements in the surface zone correlation at the 95% level, while root-zone improvements were only found for USCRN stations located in regions favourable for soil moisture DA. Our results indicate that land DA of SMAP-L2 data can account for random errors in the atmospheric forcing dataset. We base this statement on the fact that our EnKF and EnOI analysis using MERRA-2 forcing; both had comparable skill to the eOL run using NLDAS atmospheric forcing data. This could be important for regions of the world where high-quality meteorological forcing and observation data are missing, such as over the high northern latitudes and Africa.

Finally, we assimilated the ESA CCI, SMAP and SMOS SM products using the EnOI and MERRA-2 atmospheric forcing. Here, we wanted to evaluate the satellite impact on the land DA analysis. We computed surface and root-zone correlations between the three analysis products and the in situ SM data, over regions favourable for soil moisture DA. We found that assimilation of the SMAP-L2 product had the highest correlation skill, 0.69 for the surface and 0.68 for the root-zone SM. The SMOS-L2 product had the second highest overall correlation skill, 0.67 for the surface and 0.66 for the root-zone SM. While the ESA CCI product had the lowest correlation skill, 0.65 for the surface and 0.66 for the root-zone.

Land DA of SMAP data were found to perform better than assimilation of SMOS and ESA CCI data. The differences between the SMAP and SMOS analysis skill were minor, and our analysis results support other studies indicating that SMAP SM data perform slightly better than SMOS SM data when compared to in situ stations [74].

In general, our work supports other studies showing that assimilation of surface SM observations from satellites could improve the surface zone SM in land surface models [8,12,17,23,77]. It also agrees well with the same studies showing the difficulty of getting a statistically significant improvement in the root-zone SM. While the improvements in root-zone skill were minor, we did see that, using SMAP-L2 data, in combination with the MERRA-2 forcing improved the surface zone skill, even when using the computationally cheaper EnOI (in a statistically significantly manner). This result highlights the potential of the EnOI and SMAP-L2 combination for improving estimates of surface soil moisture. However, more work is needed on how to propagate this information into the root-zone.

**Author Contributions:** J.B., P.D.H., W.A.L., L.B. and C.A. conceived and designed the data assimilation experiments. D.F., P.D.H., L.B. and C.A. contributed to setting up the land surface model and the data assimilation system. J.B. conducted the experiments, did the analysis and wrote the manuscript. All authors provided comments on the manuscript.

**Funding:** The main author is supported by the Research Council of Norway (NFR PhD-grant 239947, 2015–2019).

**Acknowledgments:** The main author would like to thank Gabriëlle De Lannoy, Eric Wood and Ming Pan for valuable discussions in the construction of this paper. We would also like to thank Sam-Erik Walker for his contributions to the development of the EnKF code. Finally, we would like to thank four anonymous reviewers and the editor for their help in improving this paper.

**Conflicts of Interest:** The authors declare no conflict of interest.

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
