# Peer review of "An Evaluation of the EnKF vs. EnOI and the Assimilation of SMAP, SMOS and ESA CCI Soil Moisture Data over the Contiguous US"

_remotesensing, doi:10.3390/rs11050478_

Round 1

Reviewer 1 Report

Authors did a wonderful work to compare the performance of EnKF and EnOI over US. The result demonstrates that the combination of MERRA2, EnOI and SMAP could improve soil moisture simulation significantly. It is meaningful for the community. However, I think this manuscript could be improved in following parts:

1.     the title, current title could not reflect the main contribution or the unique of this study. It is recommended to be more specified.

2.     The presentation can be future improved. I think the finding of “the combination of MERRA2, EnOI and SMAP could improve the SSM estimation” is of the most important one, should be emphasized in abstract and also in main text.

3.     Some language related issues

a)      L18-19, it is not clear to discuss the “bias-corrected atmospheric field” here

b)      L31, partitioning…….. water available ?

c)       L216 “a close as possible”?

Author Response

Dear Reviewer, 

We would like to thank you for your helpful comments on our manuscript. A detailed response is attached. 

Sincerely,

The corresponding author. 

Reviewer 2 Report

The manuscript by Blyverket et al., 2019 presents an interesting study about data assimilation of soil moisture into land surface model. It is an important research filed and there are arising interests on improvement of model simulated soil moisture via assimilation of satellite-based soil moisture. In general, the MS is well designed and well written. I think it is worth to be published after minor revision. Below is my specific comments.

1. One general comment is regarding the bias correction of satellite-based soil moisture. The CDF matching is applied in the current study. I am wondering if the bias correction methods can have large impacts on the DA results, for example, the triple collocation method.

2.  As you described, the EnOI has advantage of less computational cost and more feasible for operational applications over large domains. I am wondering if the EnOI scheme is already applied for any operational application? Will your DA scheme be available for public?

3. Regarding the satellite-based soil moisture, I understand that the ESA CCI combined product was used. However, the lines 132-134 are confusing. I would suggest to change to: “For this reason we apply the ESA CCI combined product, which……”

4. Equation in line 267: Is it correct? It seems item Ba is missing. Please double check.

Author Response

@page { margin: 0.79in } p { margin-bottom: 0.1in; line-height: 120% }

Dear Reviewer,

We would like to thank you for your helpful comments on our manuscript. A detailed response is attached.

Sincerely,

The corresponding author.

Reviewer 3 Report

Dear Authors,

This is an excellent work making big progres in SM remote sensing data assimilation. I appreciate a great and huge work done, knowing that some minor questions are still open. Just one remark: why COSMOS network was not used for validation purposes? This network is dense in US, and  recently a big progress in employing cosmic-ray neutrons technique for validation SM remote sensing data was done. Some recent examples I listed below, but I have no doubt that Authors are aware both the advantages of new method as well as the reasons they did not use it for validation.

Andreasen, M., Jensen, K.H., Desilets, D., Franz, T.E., Zreda, M., Bogena, H.R., Looms, M.C.,2017. Status and Perspectives on the Cosmic-Ray Neutron Method for Soil Moisture Estimation and Other Environmental Science Applications. Vadose Zo. J. 16, 0.https://doi.org/10.2136/vzj2017.04.0086

Baatz, R., Bogena, H.R., Hendricks Franssen, H.J., Huisman, J.A., Qu, W., Montzka, C.,Vereecken, H., 2014. Calibration of a catchment scale cosmic-ray probe network: A comparison of three parameterization methods. J. Hydrol. 516, 231–244. https://doi.org/10.1016/j.jhydrol.2014.02.026

Kędzior, M., and J. Zawadzki. 2016. Comparative study of soil moisture

estimations from SMOS satellite mission, GLDAS database, and cosmic-

ray neutrons measurements at COSMOS station in eastern Poland.

Geoderma 283:21–31. doi:10.1016/j.geoderma.2016.07.023

However for many readers especially latter issue could be not so obvious, the more that this technique gives spatial SM measurements, not point ones. Maybe it would be advantageous for the paper to make in Introduction section some clarifying comment (or even mention) why  the cosmic-ray neutrons technique was not used for validation in this work

I leave this to your decision.

Sincerely yours,

Reviewer.

Author Response

(The authors gave the same response as above.)

Reviewer 4 Report

This work has extensively explored the data assimilation of satellite-based surface soil moisture for improving model simulated surface and root-zone soil moisture, including multiple satellite SM products. Several data assimilation issues are attempted for EnKF and EnOI methods. This reviewer only have some minor comments.

1. The authors have chosen a relatively small ensemble size as 12, please specify the reason behind this.

2. As the observation errors described in lines 341-348, do DA experiments of SMAP, SMOS, and ESA CCI have same observation errors for each grid? It is generally known that SMAP has a better performance.

3. There are some typos and grammar errors, e.g., line 34: skilful, line 39: produces.

Author Response

(The authors gave the same response as above.)
